# Stochastic series expansion quantum Monte Carlo for Rydberg arrays

**Ejaaz Merali, Isaac J. S. De Vlugt and Roger G. Melko***

**1** Department of Physics and Astronomy, University of Waterloo, Waterloo, ON, Canada
**2** Perimeter Institute for Theoretical Physics, Waterloo, ON, Canada

* rgmelko@uwaterloo.ca ,

## Abstract

Arrays of Rydberg atoms are a powerful platform to realize strongly-interacting quantum many-body systems. A common Rydberg Hamiltonian is free of the sign problem, meaning that its equilibrium properties are amenable to efficient simulation by quantum Monte Carlo (QMC). In this paper, we develop a Stochastic Series Expansion QMC algorithm for Rydberg atoms interacting on arbitrary lattices. We describe a cluster update that allows for the efficient sampling and calculation of physical observables for typical experimental parameters, and show that the algorithm can reproduce experimental results on large Rydberg arrays in one and two dimensions.

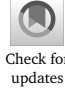

# 1  Introduction

Arrays of neutral atoms provide one of the most coherent and well-controlled experimental quantum many-body platforms available today [1, 2]. In a typical experiment, individual atoms, such as rubidium, can be trapped by laser light and driven to transition between their ground state and a *Rydberg* state: an atomic state with a large principal quantum number. With the use of optical tweezers, multiple such atoms, called Rydberg atoms, can be manipulated into arrays or lattices. Within an array, Rydberg atoms separated by a distance $R_{ij}$ (typically a few micrometers or less) experience a dipole-dipole interaction. The power-law decay of this interaction depends on how pairs of Rydberg atoms are experimentally prepared [2]; it is common to prepare pairs such that a $1/R_{ij}^6$ van der Waals (VDW) interaction is the leading-order behaviour. The resulting VDW interactions penalize the simultaneous excitation of two atoms in close proximity to each other. This effect, called the Rydberg blockade [3–5], results in a strongly-interacting Hamiltonian that can be tuned with a high degree of control to realize a variety of lattices of interest to condensed matter and quantum information physicists [6, 7].

Experimental studies are proceeding rapidly, demonstrating the creation of novel phases and phase transitions in lattice Hamiltonians in one [8] and two dimensions [9]. Theoretical studies have shown that Rydberg arrays are capable of realizing extremely rich ground state phase diagrams [10–13]. Numerical techniques have played a critical role in this theoretical exploration, providing evidence of the existence of a number of compelling phenomena, including novel quantum critical points [14, 15], floating phases [16, 17], and topologically ordered spin liquid phases [18, 19]. For these reasons, we are interested in developing a quantum Monte Carlo (QMC) algorithm for the most common Rydberg Hamiltonian. Based on the Stochastic Series Expansion (SSE) framework pioneered by Sandvik [20, 21], our algorithm provides a starting point for the exploration of a wide variety of equilibrium statistical phenomena in Rydberg arrays using this powerful and efficient QMC method.

The Hamiltonian that we consider acts on the two electronic levels of each atom $i \in \{1, 2, \ldots, N\}$: the ground state $\left|g\right\rangle \equiv \left|0\right\rangle$ and a Rydberg state $\left|r\right\rangle \equiv \left|1\right\rangle$. The Hamiltonian can be written as

$$\hat{H} = \frac{\Omega}{2} \sum_{i=1}^{N} \hat{\sigma}_i^x - \delta \sum_{i=1}^{N} \hat{n}_i + \sum_{i<j} V_{ij} \hat{n}_i \hat{n}_j, \tag{1}$$

where $N$ is the total number of Rydberg atoms. Here, the natural computational basis is the Rydberg state occupation basis, which is defined by the eigenstates of the occupation operator $\hat{n}_i = \left|1\right\rangle\!\left\langle 1\right|_i$. The eigenequations are $\hat{n}_i \left|0\right\rangle_j = 0$ for all $i, j$, and $\hat{n}_i \left|1\right\rangle_j = \delta_{i,j} \left|1\right\rangle_j$. We define $\hat{\sigma}_i^x = \left|0\right\rangle\!\left\langle 1\right|_i + \left|1\right\rangle\!\left\langle 0\right|_i$ which is an off-diagonal operator in this basis. Physically, the parameter $\Omega$ that couples to $\hat{\sigma}_i^x$ is the Rabi frequency which quantifies the atomic ground state and Rydberg state energy difference, and $\delta$ is the laser detuning which acts as a longitudinal field. As mentioned previously, a pair of atoms which are both excited into Rydberg states will

experience a VDW interaction decaying as

$$V_{ij} = \Omega \left( \frac{R_b}{r_{ij}} \right)^6. \tag{2}$$

Here $r_{ij} = (\boldsymbol{x}_i - \boldsymbol{x}_j)/a$ is the distance between the atoms, which is controlled in the experiment by tuning the lattice spacing $a$. $R_b$ is called the blockade radius, and we treat $R_b/a$ as a free parameter in the simulations below with $a = 1$. The blockade mechanism, which penalizes simultaneous excitation of atoms within the blockade radius, results in a strongly-interacting quantum Hamiltonian that produces a plethora of rich phenomena on a wide variety of lattices accessible to current and near-term experiments.

In this paper, we develop an SSE QMC implementation for the Hamiltonian Eq. (1). The remaining sections of this paper are organized as follows. In Sec. 2, we give a brief overview of the SSE framework. In Sec. 3, our SSE framework as it applies to the Hamiltonian in Eq. (1) is outlined for finite-temperature and ground state simulations. We then show results for simulations in one and two dimensions in Sec. 4, and give concluding remarks in Sec. 5.

## 2 General SSE framework

Of the numerical tools used to study strongly-interacting systems, Quantum Monte Carlo (QMC) methods are among the most powerful. Given a Hamiltonian, equilibrium properties both at finite temperature and in the ground state may be accessible to QMC simulations of various flavors. In this work, we will focus on SSE, which is related in general to "world-line" QMC methods for lattice models. Roughly, these methods use a path integral to formally map the partition function of a $d$-dimensional quantum Hamiltonian to a $d+1$-dimensional classical statistical mechanical problem. The extra dimension can be interpreted as imaginary time, and its length as the inverse temperature $\beta = 1/T$. The most efficient world-line methods for lattice models have no systematic Trotter error [22].

The successful application of QMC to a given Hamiltonian is dependent on many factors; two of the most important are the absence of the *sign problem* [23–25], and the construction of an efficient updating scheme. The absence of a sign problem implies the existence of real and positive weights derived from wavefunction or path integral configurations. These weights can therefore be interpreted probabilistically, enabling a stochastic sampling of the $d+1$-dimensional configurations. For the purposes of this paper, we define a sign problem as the presence of one or more off-diagonal matrix elements in the Hamiltonian which are *positive* when written in the computational basis. However, if one or more off-diagonal matrix elements are positive, there may exist a *sign cure* that one can apply to the Hamiltonian without altering the physics. Consider then the Rydberg Hamiltonian defined in Eq. (1). Assuming that the Rabi frequency $\Omega > 0$, this Hamiltonian naively appears to be sign-problematic in the native Rydberg occupation basis. However, upon application of a trivial canonical transformation on each lattice site (discussed further in Sec. 3), the sign of this off-diagonal term can be flipped without affecting the physics of the system.

The second condition required for a successful QMC algorithm is the construction of an efficient updating scheme. This can be a highly non-trivial endeavour, which ultimately affects the accessible lattice sizes of the QMC simulation. General concepts often guide the design of efficient QMC update algorithms, such as the construction of *cluster* updates that are non-local in space and/or imaginary time akin to the loop or worm algorithms [26–28]. However, the specific design and performance of an update algorithm depends crucially on the flavor of QMC.

In the sections below, we detail an algorithm for simulating Rydberg Hamiltonians based on the SSE method [20, 21, 28–31]. Our algorithm follows Sandvik's development of the spin-1/2 transverse-field Ising model [32], generalized to the Rydberg Hamiltonian Eq. (1). In this section, we offer only a brief review of the general SSE formalisms for finite- and zero-temperature QMC simulations, which are covered extensively in the literature.

## 2.1 Finite temperature formalism

The finite-temperature SSE method is based on the Taylor series expansion of the partition function in a computational basis $\{|\alpha_0\rangle\}$ – for example, the $S^z$ basis for a spin-1/2 system, or the Rydberg occupation basis. By explicitly writing out the trace, the partition function becomes,

$$Z = \text{Tr}\left\{e^{-\beta\hat{H}}\right\} = \sum_{\alpha_0} \left\langle\alpha_0\right| \sum_{n=0}^{\infty} \frac{\beta^n}{n!}(-\hat{H})^n \left|\alpha_0\right\rangle \tag{3a}$$

$$= \sum_{\{\alpha_p\}} \sum_{n=0}^{\infty} \frac{\beta^n}{n!} \prod_{p=1}^{n} \left\langle\alpha_{p-1}\right| -\hat{H} \left|\alpha_p\right\rangle, \tag{3b}$$

where $\beta$ is the inverse temperature, $\hat{H}$ is the Hamiltonian, and in Eq. (3b) we've inserted a resolution of the identity in terms of the basis states $\{|\alpha_p\rangle\}$ between each product of $-\hat{H}$. It's at this point where the mapping to a $d+1$-dimensional classical problem is apparent, where the additional imaginary time direction comes from the expansion order $n$, and the subscripts on the basis states $|\alpha_p\rangle$ enumerate the location in imaginary time. Crucially, translational invariance along this dimension is enforced by the trace, i.e. $|\alpha_n\rangle = |\alpha_0\rangle$.

We proceed from Eq. (3b) by decomposing the Hamiltonian into elementary lattice operators,

$$\hat{H} = -\sum_{t,a} \hat{H}_{t,a}, \tag{4}$$

where we use the label $t$ to refer to the operator "type" (e.g. whether $\hat{H}_{t,a}$ is diagonal or off-diagonal) and the label $a$ to denote the lattice unit that $\hat{H}_{t,a}$ acts on. In the implementation of SSE QMC, one has a large amount of freedom to decide the basic lattice units that make up this lattice decomposition (e.g. a site, bond, plaquette, etc). From this, the partition function can be written as

$$Z = \sum_{\{\alpha_p\}} \sum_{n=0}^{\infty} \sum_{S_n} \frac{\beta^n}{n!} \prod_{p=1}^{n} \left\langle\alpha_{p-1}\right| \hat{H}_{t_p,a_p} \left|\alpha_p\right\rangle, \tag{5}$$

where $S_n$ represents a particular sequence of $n$ elementary operators in imaginary time, $S_n = [t_1, a_1], [t_2, a_2], \cdots, [t_n, a_n]$. In other words, for a given sequence $S_n$, the basis state $|\alpha_{p-1}\rangle$ is propagated in the imaginary time direction to another basis state $|\alpha_p\rangle$ by the elementary operator $\hat{H}_{t_p,a_p}$ as

$$\hat{H}_{t_p,a_p} \left|\alpha_{p-1}\right\rangle \propto \left|\alpha_p\right\rangle. \tag{6}$$

With the representation of the partition function in Eq. (5), the SSE configuration space is defined by $S_n$, the basis states $|\alpha_p\rangle$, and the expansion order $n$. We thus see that each matrix element of $\hat{H}_{t,a}$ must be positive so as to avoid the sign problem, ensuring that each SSE configuration can be interpreted as a probabilistic weight.

The partition function in Eq. (5) is still not suitable for numerical implementation due to the infinite sum over the expansion order $n$. By observing that the distribution of $n$ which contribute to the partition function always has a range bounded by some $n_{\max}$ [33], a maximum

imaginary time length $M$ can be automatically chosen during the equilibration phase of the QMC. Enforcing $M > n_{\max}$, the actual expansion order $n$ is allowed to fluctuate during the QMC simulation. Given that $M > n$, $M - n$ slices in imaginary time will have trivial identity operators. Typically, $M$ is grown until the fraction of non-identity operators $n/M$ present in the operator sequence is greater than 80%.

In a given simulation, we place the identity matrix $\mathbb{I}$ at $M - n$ positions. Accounting for all the possible placements, we arrive at the final expression for the partition function,

$$Z = \sum_{\{\alpha_p\}} \sum_{S_M} \frac{\beta^n (M-n)!}{M!} \prod_{p=1}^{M} \left\langle \alpha_{p-1} \middle| \hat{H}_{t_p, a_p} \middle| \alpha_p \right\rangle = \sum_{\{\alpha_p\}} \sum_{S_M} \Phi(\{\alpha_p\}, S_M), \tag{7}$$

where $S_M$ is a new operator sequence that includes the sum over $n$, and $\Phi(\{\alpha_p\}, S_M)$ is the generalized SSE configuration space weight. The $\beta$-dependence is implied throughout. We are now free to devise update procedures to produce a Markov Chain in the configuration space labelled by the basis states $\{|\alpha_p\rangle\}$ and the elementary operator string $S_M$. We defer an explanation of possible update procedures to those specifically used in our SSE implementation for Rydberg atoms in Sec. 3.

## 2.2 Ground state projector formalism

Formally, the zero-temperature SSE method is based around a projector QMC representation. One can write an arbitrary trial state $|\alpha_r\rangle \in \{|\alpha\rangle\}$ (the computational basis) in terms of the eigenstates $\{|\lambda_m\rangle, m \in 0, 1, \cdots, \mathcal{D}-1\}$ of the Hamiltonian with Hilbert space dimension $\mathcal{D}$ as $|\alpha_r\rangle = \sum_{m=0}^{\mathcal{D}-1} c_m |\lambda_m\rangle$. The ground state $|\lambda_0\rangle$ can then be projected out of $|\alpha_r\rangle$ via

$$(-\hat{H})^M |\alpha_r\rangle = c_0 |E_0|^M \left( |\lambda_0\rangle + \sum_{m=1}^{\mathcal{D}-1} \frac{c_m}{c_0} \left( \frac{E_m}{E_0} \right)^M |\lambda_m\rangle \right) \xrightarrow{M \to \infty} c_0 |E_0|^M |\lambda_0\rangle, \tag{8}$$

where we've assumed that appropriate shifts to the Hamiltonian have been done so as to make $E_0$ the largest (in magnitude) eigenvalue of $H$.

The normalization factor that we now need to devise an importance sampling procedure for is

$$Z \equiv \left\langle \lambda_0 \middle| \lambda_0 \right\rangle = \left\langle \alpha_\ell \middle| (-\hat{H})^M (-\hat{H})^M \middle| \alpha_r \right\rangle, \tag{9}$$

for sufficiently large "projector length" $2M$. Here, the trial states $|\alpha_\ell\rangle, |\alpha_r\rangle \in \{|\alpha\rangle\}$ need not be equal, $|\alpha_\ell\rangle \neq |\alpha_r\rangle$, breaking translational invariance in imaginary time. As before with the finite-temperature SSE method, we insert resolutions of the identity in terms of the computational basis states $\{|\alpha\rangle\}$ in between each product of $-\hat{H}$ and then decompose our Hamiltonian as in Eq. (4) to arrive at the following representation of the normalization:

$$Z = \sum_{\{\alpha_p\}} \sum_{S_M} \prod_{p=1}^{2M} \left\langle \alpha_{p-1} \middle| \hat{H}_{t_p, a_p} \middle| \alpha_p \right\rangle, \tag{10}$$

where $|\alpha_0\rangle \equiv |\alpha_\ell\rangle$, $|\alpha_{2M}\rangle \equiv |\alpha_r\rangle$, the subscript $p$ denotes the imaginary time location, and $S_M$ denotes a particular sequence of elementary operators similar to the finite-temperature case. As before, $\hat{H}_{t_p, a_p}$ propagates the computational basis states according to Eq. (6), each matrix element of $\hat{H}_{t,a}$ must be positive to avoid sign problems, and the configuration space to be importance-sampled is the combination of $\{|\alpha_p\rangle\}$ and $S_M$.

## 2.3 Observables

Estimators for various diagonal and off-diagonal observables can be calculated with a variety of procedures in SSE, and with some notable exceptions their derivations are mostly beyond the scope of this paper. Diagonal observables can be trivially calculated directly from samples in the basis $\{|\alpha\rangle\}$. Quite generally, for finite-temperature, the simulation cell can be sampled at any point in imaginary time, while at zero-temperature one must sample at the middle of the simulation cell due to the structure of the projection framework. Many off-diagonal observables can also be efficiently calculated in finite-temperature formalism. For example, since we have access to a compact expression for the partition function (Eq. (5)), one may take suitable derivatives of this expression to extract thermodynamic quantities such as the energy,

$$E = -\frac{\partial \ln Z}{\partial \beta} = -\frac{\langle n \rangle}{\beta}. \tag{11}$$

Note that in the zero-temperature SSE framework, expressions for off-diagonal observables such as the energy may be very different. In general, the observable $\hat{A}$ can be calculated as

$$\langle \hat{A} \rangle = \frac{\langle \lambda_0 | \hat{A} | \lambda_0 \rangle}{\langle \lambda_0 | \lambda_0 \rangle} = \frac{\langle \alpha_\ell | (-\hat{H}^M) \hat{A} (-\hat{H}^M) | \alpha_r \rangle}{\langle \alpha_\ell | (-\hat{H}^M)(-\hat{H}^M) | \alpha_r \rangle}. \tag{12}$$

The non-triviality of calculating general observables $\hat{A}$ in terms of SSE simulation parameters is evident from this. By inserting $\hat{A} = \hat{H}$ into this expression, we offer a derivation for the ground state energy for the Rydberg Hamiltonian SSE in Sec. 3.3.

## 3 SSE implementation for Rydberg atoms

The previous section presented some generalities of the SSE framework in both the finite-temperature and ground state projector formalisms. To translate these formalisms into simulating the Rydberg Hamiltonian Eq. (1), we must define the basis states $\{|\alpha\rangle\}$, elementary lattice operators in Eq. (4), and the update strategy. Naturally, the choice of computational basis is that of the Rydberg occupation basis: $\{|\alpha\rangle\} = \{\bigotimes_{i=1}^N |n_i\rangle, n_i = 0, 1\}$.

To make progress on defining the elementary lattice operators in Eq. (4), as well as the update strategy, the specific form of the Hamiltonian must be considered. The Rydberg Hamiltonian Eq. (1) takes the form of a quantum Ising model with transverse and longitudinal fields. Since the transverse-field term is positive in the Rydberg occupation basis, we must devise a sign cure. Consider a unitary transformation $\hat{U} = \bigotimes_{i=1}^N \hat{\sigma}_i^z = \bigotimes_{i=1}^N [\mathbb{I} - 2\hat{n}_i]$. Altogether, the Hamiltonian Eq. (1) is unitarily transformed to

$$\hat{U}^\dagger \hat{H} \hat{U} = -\frac{\Omega}{2} \sum_{i=1}^N \hat{\sigma}_i^x - \delta \sum_{i=1}^N \hat{n}_i + \sum_{i<j} V_{ij} \hat{n}_i \hat{n}_j, \tag{13}$$

which is now free of a sign problem in the native Rydberg occupation basis.

Now that the sign problem has been alleviated, we can proceed with writing the Hamiltonian Eq. (13) in the form of Eq. (4). Motivated by Refs. [28, 32], we define the elementary lattice operators of the SSE as

$$\hat{H}_{0,0} = \mathbb{I}, \tag{14a}$$

$$\hat{H}_{-1,a} = \frac{\Omega}{2} \hat{\sigma}_i^x, \tag{14b}$$

$$\hat{H}_{1,a} = \frac{\Omega}{2} \mathbb{I}, \tag{14c}$$

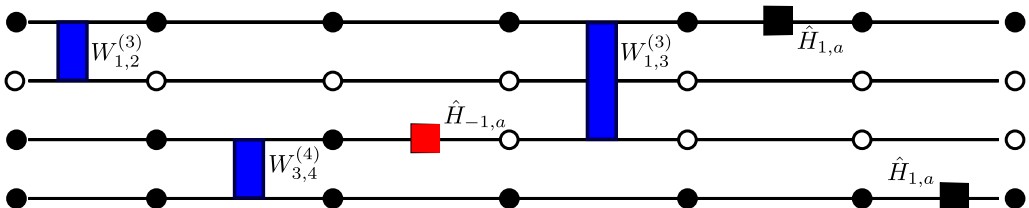

Figure 1: A ground state projector SSE simulation cell example of the SSE operator breakup in Eq. (14) with matrix elements in Eq. (15) for $2M = 6$. Rydberg occupations labelled with a filled (unfilled) circle denote $n_i = 1(0)$. The occupation configuration on the left is $\langle \alpha_\ell |$, and on the right is $| \alpha_r \rangle$.

and

$$\hat{H}_{1,b} = -V_{ij}\hat{n}_i\hat{n}_j + \delta_b(\hat{n}_i + \hat{n}_j) + C_{ij}. \tag{14d}$$

Here $\delta_b = \delta/(N-1)$ is the reduced detuning parameter since the sum $\delta \sum_i \hat{n}_i$ has been moved into the sum over pairs $\sum_{i<j}$, and $C_{ij} = |\min(0, \delta_b, 2\delta_b - V_{ij})| + \varepsilon |\min(\delta_b, 2\delta_b - V_{ij})|$ is added to $\hat{H}_{1,b}$ so that all of its matrix elements remain non-negative, where $\varepsilon \geq 0$. Note that Eq. (14a) is only used for finite temperature simulations. The additional $\varepsilon$ term in the definition of $C_{ij}$ is typically employed to aid numerics [28]. In contrast to Ref. [28], we define $\varepsilon$ as a multiplicative constant as opposed to an additive one, since the different $C_{ij}$s vary greatly in magnitude.

It is helpful to show the matrix elements of each of these local operators since these values are the foundation of importance sampling for each of the local operators. The matrix elements in the Rydberg occupation basis are

$$\langle 1|\hat{H}_{-1,a}|0\rangle = \langle 0|\hat{H}_{-1,a}|1\rangle = \frac{\Omega}{2}, \tag{15a}$$

$$\langle 1|\hat{H}_{1,a}|1\rangle = \langle 0|\hat{H}_{1,a}|0\rangle = \frac{\Omega}{2}, \tag{15b}$$

$$W_{ij}^{(1)} \equiv \langle 00|\hat{H}_{1,b}|00\rangle = C_{ij}, \tag{15c}$$

$$W_{ij}^{(2)} \equiv \langle 01|\hat{H}_{1,b}|01\rangle = \delta_b + C_{ij}, \tag{15d}$$

$$W_{ij}^{(3)} \equiv \langle 10|\hat{H}_{1,b}|10\rangle = \delta_b + C_{ij}, \tag{15e}$$

and

$$W_{ij}^{(4)} \equiv \langle 11|\hat{H}_{1,b}|11\rangle = -V_{ij} + 2\delta_b + C_{ij}, \tag{15f}$$

where subscripts $i, j$ on matrix elements $W_{ij}^{(1,2,3,4)}$ here contain the spatial location dependence. Fig. 1 shows an example of a zero-temperature SSE simulation cell of such an operator breakup. A finite temperature simulation cell would look very similar, except translational invariance in imaginary time forces the Rydberg occupation configurations on the left and right edges to be the same.

## 3.1 Diagonal update

Updates to the $d + 1$-dimensional configurations in the SSE QMC framework typically occur via a number of separate steps – most importantly a *diagonal update* followed by a non-local *cluster update* (often called an *off-diagonal update*). In the diagonal update, the algorithm searches through every imaginary time slice in the SSE simulation cell to propose adding or

removing diagonal operators. Note, a proposal to remove a diagonal operator without a replacement occurs only in the finite-temperature formalism. In this way the topology of the simulation cell is changed by altering the sequence of operators $S_M$ without altering the world lines of each atom. Below, we outline the finite- and zero-temperature diagonal updates in the next two sections for the elementary operator breakdown outlined in above for the Rydberg Hamiltonian.

### 3.1.1 Finite temperature

For the finite-temperature simulation cell defined by Eq. (7), the diagonal update proceeds by looping through every imaginary time slice $p \in \{1, 2, \cdots, M\}$ and attempting the following steps at each.

1. If $\hat{H}_{1,a}$ or $\hat{H}_{1,b}$ is encountered, remove it ($n \to n-1$) with probability

$$A([1,a]_p \text{ or } [1,b]_p \to [0,0]_p) = \min\left(\frac{M-n+1}{\beta\mathcal{N}}, 1\right), \tag{16}$$

   where $\mathcal{N}$ is a normalizing constant which we will define below.

2. If $\hat{H}_{0,0}$ is encountered, decide whether or not to attempt inserting $\hat{H}_{1,a}$ or $\hat{H}_{1,b}$ ($n \to n+1$) with the probability

$$A([0,0]_p \to [1,a]_p \text{ or } [1,b]_p) = \min\left(\frac{\beta\mathcal{N}}{M-n}, 1\right). \tag{17}$$

3. If it was decided to attempt inserting $\hat{H}_{1,a}$ or $\hat{H}_{1,b}$ in the previous step, we choose $\hat{H}_{1,a}$ at site $i$ or $\hat{H}_{1,b}$ at bond $(i,j)$ by sampling the (unnormalized) probability distribution

$$P_{ij} = \begin{cases} \frac{\Omega}{2} & i = j \\ \max\left(W_{ij}^{(1)}, W_{ij}^{(2)}, W_{ij}^{(3)}, W_{ij}^{(4)}\right) & i \neq j \end{cases}. \tag{18}$$

   We call the normalizing constant of this distribution $\mathcal{N} = \sum_{ij} P_{ij}$. We employ the Alias method [34–36] to draw samples from this distribution in $\mathcal{O}(1)$ time. Sampling this gives an operator corresponding to the given matrix element (Eq. (15)) whose insertion will be attempted at the spatial location $(i,j)$ in the current imaginary time slice $p$ ($\hat{H}_{1,a}$ if $i = j$ or $H_{1,b}$ if $i \neq j$). If $\hat{H}_{1,a}$ is chosen, its insertion at site $i$ is accepted. If $\hat{H}_{1,b}$ is chosen, one of two things may happen. The configuration at the current imaginary time slice $p$ is given by $\left|n_{1,p}, n_{2,p}, \cdots, n_{N,p}\right\rangle$. If $\left|n_{i,p}, n_{j,p}\right\rangle$ matches the sampled matrix element of $\hat{H}_{1,b}$ (i.e. one of $W_{ij}^{(1)}, W_{ij}^{(2)}, W_{ij}^{(3)}$, or $W_{ij}^{(4)}$), the insertion is accepted. Otherwise, the insertion of $\hat{H}_{1,b}$ at location $(i,j)$ is accepted with probability

$$\frac{W_{ij}^{(\text{actual})}}{W_{ij}^{(\text{sampled})}}, \tag{19}$$

   where $W_{ij}^{(\text{actual})} = \left\langle n_{i,p}, n_{j,p}\left|\hat{H}_{1,b}\right|n_{i,p}, n_{j,p}\right\rangle$ and $W_{ij}^{(\text{sampled})}$ was sampled from the distribution Eq. (18).

4. If $\hat{H}_{-1,a}$ is encountered, propagate the state: $\left|n_p\right\rangle \propto \hat{H}_{-1,a}\left|n_{p-1}\right\rangle$.

5. Repeat step 1 at the next imaginary time slice.

Note, there are different ways of attempting to insert diagonal operators than what is outlined in step #3. However, as suggested by Sandvik [32], what is depicted in step #3 is the most efficient way to sample non-uniform diagonal operator matrix elements. We offer a formal reasoning for this statement along with derivations of Eqs. (16) and (17) in the Appendix.

### 3.1.2 Ground state projector

The simulation cell for the $T = 0$ ground state projector version of the SSE is given by Eq. (10). Since the projector length $M$ is not allowed to fluctuate, we do not pad $S_M$ with identity operators $\hat{H}_{0,0}$. Therefore, for every imaginary time slice in the diagonal update, one always removes the current diagonal operator and continues attempting to insert a new diagonal operator until one of the attempts is successful. This amounts to repeating step #3 in Sec. 3.1.1 at every imaginary time slice until a successful insertion is achieved. Another way to see this is by taking the $\beta \to \infty$ limit of the insertion (Eq. (17)) and removal (Eq. (16)) probabilities. As before, when $\hat{H}_{-1,a}$ is encountered, we simply propagate the state then continue on to the next imaginary time slice.

## 3.2 Cluster updates

The diagonal update procedures in Sec. 3.1 allow for new diagonal operators to replace current ones. However these updates alone are not ergodic, as they clearly do not sample operators $\hat{H}_{-1,a}$, i.e. they do not alter the world line configurations. Thus, each diagonal update in the SSE is followed by a non-local cluster update. To devise an ergodic algorithm for the Rydberg Hamiltonian, we use the cluster update devised by Sandvik called the *multibranch* cluster update, which is described in Refs. [31, 32, 37]. This is a highly non-local update originally designed for the SSE implementation of the transverse-field Ising model. We offer a brief explanation of this cluster update in the following paragraph.

Switching to a graph-based vocabulary, one may think of matrix elements in Eq. (15) as *vertices* in a graph. Vertices from the elementary bond operator $\hat{H}_{1,b}$ comprise of four *legs* (two Rydberg occupation states from the ket and bra), while vertices from site operators $\hat{H}_{1,a}$ and $\hat{H}_{-1,a}$ have two legs (one Rydberg occupation state from the ket and bra). The operators $H_{0,0}$ in the finite-temperature case are ignored. Multibranch clusters are formed by beginning at one of the legs of a random site operator vertex and traversing away from this operator in the imaginary time direction. If a bond vertex is encountered, all four vertex legs are added to the cluster and the cluster continues to grow by branching out of all three remaining exit legs. If a site vertex is encountered, the cluster terminates at that newly encountered leg. For finite-temperature simulations, if the edge of the simulation is reached by the cluster, it must loop around to the opposite edge in order to respect periodic boundary conditions in imaginary time. If the edge of the simulation is reached in a ground state projector simulation, the cluster terminates at the boundary edge.

Fig. 2 shows an example of a ground state projector SSE simulation cell wherein a multibranch cluster is pictured by the green region. Updating clusters consists of flipping all legs (Rydberg occupations) and vertex types that are within the cluster in a corresponding fashion. Since cluster weights may change when flipping, detailed balance must be satisfied by flipping clusters with the Metropolis probability

$$P_{\text{flip}} = \min\left(1, \frac{\mathcal{W}'}{\mathcal{W}}\right), \tag{20}$$

where $\mathcal{W}$ is weight of the cluster defined as the product of vertices $v_i$ (matrix elements with

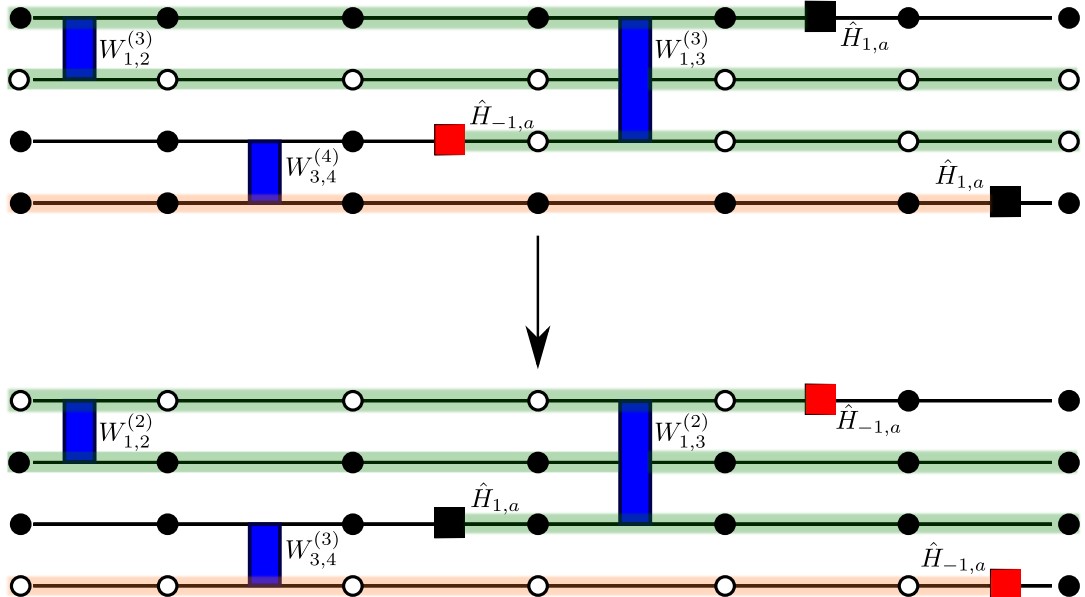

Figure 2: A ground state projector SSE simulation cell example of the SSE operator breakup in Eq. (14) with matrix elements in Eq. (15) for $2M = 6$. Rydberg occupations labelled with a filled (unfilled) circle denote $n_i = 1(0)$, with $\langle \alpha_\ell |$ on the left edge and $|\alpha_r \rangle$ on the right edge. In the upper simulation cell, we show examples of the multibranch (green) and line (orange) clusters. These clusters are probabilistically flipped according Eq. (20). If each cluster pictured here is flipped, the lower simulation cell is what results.

values $W(v_i)$ found in Eq. (15)) belonging to the cluster $c$:

$$\mathcal{W} = \prod_{v_i \in c} W(v_i). \tag{21}$$

$\mathcal{W}'$ denotes the weight of cluster $c$ from flipping it, therefore changing the vertex types $v_i \in c$. For instance, the upper pane of Fig. 2 shows a multibranch cluster (green) that has a weight $\mathcal{W} \propto W_{1,2}^{(3)} \times W_{1,3}^{(3)} \times \langle 1|\hat{H}_{1,a}|1\rangle \times \langle 1|\hat{H}_{-1,a}|0\rangle$. When flipped (lower pane), it has a weight $\mathcal{W}' \propto W_{1,2}^{(2)} \times W_{1,3}^{(2)} \times \langle 0|\hat{H}_{-1,a}|1\rangle \times \langle 1|\hat{H}_{1,a}|1\rangle$. Note that if the simulation cell's outer edge states are initialized to $\bigotimes_{i=1}^{N} \frac{1}{\sqrt{2}}(|0\rangle_i + |1\rangle_i)$ (i.e. simulation cell edge states are randomly initialized), weight changes do not manifest from flipping Rydberg occupations at the simulation cell edges. As we are now taking the weight change into account, we may need to visit every leg in a cluster twice: once to accumulate the weights and then again to flip each of these legs if the update was accepted.

The multibranch cluster works exceptionally well for the transverse-field Ising model partially owing to the fact that this update results in efficient, highly non-local configuration changes. In particular, multibranch clusters are formed deterministically and do not accrue a weight change upon flipping, allowing the update to be accepted with probability $1/2$ [32]. In the case of the Rydberg Hamiltonian Eq. (1), the presence of the laser detuning $\delta$ and the nature of the interactions $\hat{n}_i \hat{n}_j$ require that the ratio of weights in Eq. (20) must be considered for every update, though the clusters are still constructed deterministically.

Intuitively, we expect the multibranch update to be inefficient for many $R_b$ and $\delta$ combinations, as any cluster containing either the matrix element $W_{ij}^{(1)}$ or $W_{ij}^{(4)}$ will be frozen since the flipped counterpart has weight zero (or, in the case of a small non-zero $\varepsilon$, a weight

close to zero). Additionally, we expect the long-range interactions to increase the number of frozen clusters as each cluster will have a higher likelihood of containing a $W^{(1)}$ or $W^{(4)}$ matrix element. This motivates us to search for an update which instead of proposing moves $W^{(1)} \leftrightarrow W^{(4)}$, proposes moves such as $W^{(1)} \leftrightarrow W^{(2)}$, $W^{(1)} \leftrightarrow W^{(3)}$ and so on, flipping only a single spin of a bond.

From an alternative combinatorial perspective, a spatially non-local cluster like that in Fig. 2 touches $K$ physical sites and thus has (in general) $2^K$ states. Due to the $\sigma_z \rightarrow -\sigma_z$ symmetry of the transverse-field Ising model, the bond operator (after adding the constant energy shift) has only two non-zero matrix elements and the cluster therefore only has two possible configurations with non-zero weights which the multibranch update alternates between. The multibranch update is thus optimal for this case. However, in the Rydberg case most bonds have more than two non-zero weights. The multibranch update is therefore no longer sufficient to explore all $2^{O(K)}$ configurations of each cluster. This is where the line cluster update comes in.

The line cluster update is a local-in-space and non-local-in-imaginary-time cluster inspired by Ref. [38] (similar updates have also been proposed in Refs. [39,40]). Like the multibranch clusters, the line clusters also terminate on site vertices and are thus deterministically constructed. However, if an $\hat{H}_{1,b}$ vertex is encountered, only the adjacent leg in imaginary time is added to the cluster and it continues to propagate in the imaginary time direction until reaching site operators. This cluster is flipped with the same probability in Eq. (20). For instance, the orange line cluster in Fig. 2 has a weight $\mathcal{W} \propto W_{3,4}^{(4)} \times \langle 1|\hat{H}_{1,a}|1\rangle$. When flipped, it has a weight $\mathcal{W}' \propto W_{3,4}^{(3)} \times \langle 0|\hat{H}_{-1,a}|1\rangle$.

In our simulations, we define a Monte Carlo step as a diagonal update followed by a off-diagonal update in which all possible clusters are constructed and flipped independently according to the Metropolis condition. The specific type of off-diagonal update we use (line or multibranch) is selected beforehand.

### 3.3 Ground state energy estimator

Given the normalization in Eq. (9), we wish to find a compact expression for the ground state energy,

$$\langle \hat{H} \rangle = E_0 = \frac{1}{Z} \left\langle \alpha_\ell \left| (-\hat{H})^M \hat{H}(-\hat{H})^M \right| \alpha_r \right\rangle, \tag{22}$$

in terms of parameters in the SSE simulation cell. The following derivation for such an expression for $E_0$ applies to any SSE elementary operator breakup wherein one of the local operators is a multiple of the identity, which applies to our case (see Eq. (14c)). For generality, we denote such a local operator by $\hat{H}_{h\mathbb{I}} = h\mathbb{I}$.

In the $d + 1$ simulation cell, the presence of $\hat{H}_{h\mathbb{I}}$ does not alter world line paths. Therefore, in the summation over all possible operator strings $S_M$ in our normalization, operator strings that contain $m$ instances of $\hat{H}_{h\mathbb{I}}$ operators will have the same weight. If $\tilde{M} = 2M - m$ represents the operator string with all $\hat{H}_{h\mathbb{I}}$ operators removed, then

$$Z = \sum_{\{\alpha\}} \sum_{S_M} h^m \prod_{p=1}^{\tilde{M}} \left\langle \alpha_{p-1} \left| \hat{H}_{t_p,a_p} \right| \alpha_p \right\rangle,$$

where the operator sequence $S_M$ still contains the information regarding where all $m$ $\hat{H}_{h\mathbb{I}}$ operators are placed. We can take advantage of degeneracies $\binom{2M}{m}$ from imaginary time combinatorics and $N^m$ from the number of spatial locations $N$ that this operator can exist on at a

given time slice. This allows for a new configuration-space representation defined by $\{\alpha\}$, $S_{\tilde{M}}$, and $m$. In this new space,

$$Z = \sum_{\{\alpha\}} \sum_{S_{\tilde{M}}} \sum_m N^m \frac{(2M)!}{\tilde{M}!m!} h^m \prod_{p=1}^{\tilde{M}} \left\langle \alpha_{p-1} \middle| \hat{H}_{t_p,a_p} \middle| \alpha_p \right\rangle.$$

Let's now make the change of variables $q = m + 1$. Importantly, $\tilde{M}$ remains fixed, but the new projector length is $2Q = 2M + 1 = \tilde{M} + q$. After the change of variables, the normalization is

$$Z = \sum_{\{\alpha\}} \sum_{S_{\tilde{M}}} \sum_q N^{q-1} \frac{(2Q-1)!}{\tilde{M}!(q-1)!} h^{q-1} \prod_{p=1}^{\tilde{M}} \left\langle \alpha_{p-1} \middle| \hat{H}_{t_p,a_p} \middle| \alpha_p \right\rangle.$$

If we let

$$\Phi(\alpha, S_{\tilde{M}}, q) = N^q \frac{(2Q)!}{\tilde{M}!q!} h^{q-1} \prod_{p=1}^{\tilde{M}} \left\langle \alpha_{p-1} \middle| \hat{H}_{t_p,a_p} \middle| \alpha_p \right\rangle, \tag{23}$$

then

$$Z = \frac{1}{2Q \times N} \sum_{\{\alpha\}} \sum_{S_{\tilde{M}}} \sum_q \Phi(\alpha, S_{\tilde{M}}, q) q.$$

Let's now turn to evaluating $\left\langle \alpha_\ell \middle| (-\hat{H})^M \hat{H} (-\hat{H})^M \middle| \alpha_r \right\rangle$. If we insert a resolution of the identity over basis states $\{|\alpha\rangle\}$ between every product of $(-\hat{H})$ and proceed in similar fashion to deriving Eq. (10), then

$$\left\langle \alpha_\ell \middle| (-\hat{H})^M \hat{H} (-\hat{H})^M \middle| \alpha_r \right\rangle = -\sum_{\{\alpha\}} \sum_{S_M} \prod_{p=1}^{2M+1} \left\langle \alpha_{p-1} \middle| \hat{H}_{t_p,a_p} \middle| \alpha_p \right\rangle.$$

We can also take into account for degeneracies of operator strings containing $q$ instances of $H_{h\mathbb{I}}$ operators as before, giving

$$\left\langle \alpha_\ell \middle| (-\hat{H})^m \hat{H} (-\hat{H})^m \middle| \alpha_r \right\rangle = -\sum_{\{\alpha\}} \sum_{S_{\tilde{M}}} \sum_q N^q \binom{2M+1}{q} h^q \prod_{p=1}^{\tilde{M}} \left\langle \alpha_{p-1} \middle| \hat{H}_{t_p,a_p} \middle| \alpha_p \right\rangle$$

$$= -h \sum_{\{\alpha\}} \sum_{S_{\tilde{M}}} \sum_q N^q \frac{(2M+1)!}{\tilde{M}!q!} h^{q-1} \prod_{p=1}^{\tilde{M}} \left\langle \alpha_{p-1} \middle| \hat{H}_{t_p,a_p} \middle| \alpha_p \right\rangle.$$

As the above expression is already naturally working within the change of variables performed previously ($q = m + 1$, $2Q = 2M + 1$), using Eq. (23) we can write

$$\left\langle \alpha_\ell \middle| (-\hat{H})^m \hat{H} (-\hat{H})^m \middle| \alpha_r \right\rangle = -h \sum_{\{\alpha\}} \sum_{S_{\tilde{M}}} \sum_q \Phi(\alpha, S_{\tilde{M}}, q).$$

Putting everything together, we have

$$\frac{E_0}{N} = \frac{\langle \hat{H} \rangle}{N} = -2Q \frac{h}{\langle q \rangle}. \tag{24}$$

# 4 Results

Numerous recent experimental works have showcased the future potential of Rydberg atoms as a platform for quantum computation and for realizing a host of quantum many-body phenomena. Motivated in particular by the experiments of Bernien *et al.* [8] and Ebadi *et al.* [9], we present results that showcase our SSE QMC algorithm for a 51 atom one-dimensional (1D) chain and a $16 \times 16$ square array of Rydberg atoms, both with open boundary conditions. All results reported in this section take $\Omega = 1$ and $R_b = 1.2$.

## 4.1 51 atom 1D chain

At finite temperature, an SSE QMC simulation is allowed to grow in imaginary time during the equilibration phase. Therefore, a suitably-converged simulation cell size is automatically calculated during equilibration (see Sec. 2.1). Fig. 3 shows the estimated energy density, calculated using Eq. (11), and the corresponding simulation cell size $M$ for various $\delta/\Omega$ values. The line update was chosen as the cluster update for each simulation. As expected, for higher (lower) temperatures we observe that the automatically-calculated simulation cell size is smaller (larger).

Sec. 3.2 outlined the two cluster updates we have implemented for our SSE QMC algorithm. The question of which cluster update is best to employ will undoubtedly depend on $R_b$, $\delta/\Omega$, and system size. However, MC observables like the finite- (Eq. (11)) or zero-temperature (Eq. (24)) energies that strictly depend on SSE simulation-cell parameters and not the basis states $\{|\alpha\rangle\}$ are extremely robust to the choice of cluster update; the mechanics of the diagonal update are far more important since the diagonal updates do not modify $\{|\alpha\rangle\}$.

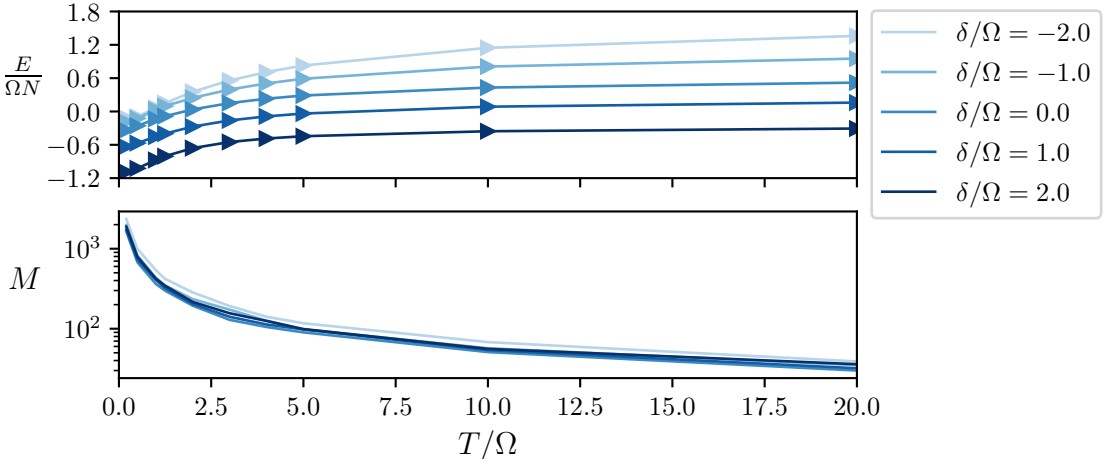

Figure 3: The estimated energy density $E/\Omega N$ and the equilibrated simulation cell size $M$ for an $N = 51$ 1D chain of Rydberg atoms with $R_b = 1.2$ as a function of temperature $T/\Omega$ and $\delta/\Omega$. Error bars in the energy density are smaller than the markers. Each data point represents an independent SSE QMC simulation (line cluster updates only – see Sec. 3.2) wherein $10^7$ successive measurements were taken and placed into 500 bins. These 500 binned measurements were then used to calculate statistics.

At zero temperature we do not automatically grow the simulation cell size / projector length $2M$ — typically, it is manually converged. For our example value of the blockade radius, $R_b = 1.2$, we consider a value of $\delta/\Omega = 1.1$ which is near a quantum phase transition (QPT)

in 1D [8]. Fig. 4 shows the estimated ground state energy, calculated using Eq. (24), versus projector lengths $2M$. The line update was chosen as the cluster update for each simulation. From this, a suitably-converged projector length $2M$ can be interpolated. We observe that $2M = 2.4 \times 10^4$ gives energies converged to well within error bars of those with larger projector lengths. We use this projector length henceforth for the 51 Rydberg atom results.

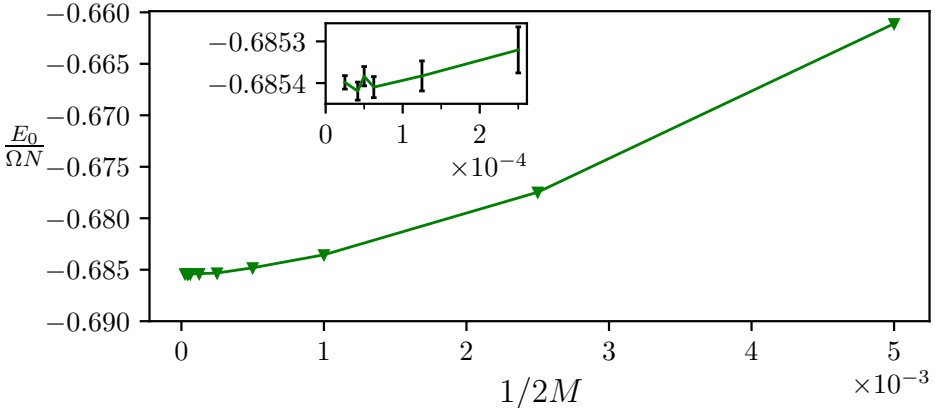

Figure 4: The estimated energy density $E_0/\Omega N$ (Eq. (24)) vs the simulation cell size $2M$ for an $N = 51$ 1D chain of Rydberg atoms with $R_b = 1.2$ and $\delta/\Omega = 1.1$ as a function of the inverse projector length $1/2M$. Each data point represents an independent SSE QMC simulation (line cluster updates only – see Sec. 3.2) wherein $10^7$ successive measurements were taken and placed into 500 bins. These 500 binned measurements were then used to calculate statistics via a standard jackknife routine. In the main plot, error bars are smaller than the plot markers.

Fig. 5 shows the estimated absolute value of the staggered magnetization,

$$|M_s| = \left| \sum_{j=1}^{N} (-1)^j \left( n_j - \frac{1}{2} \right) \right|,\tag{25}$$

where $n_j = 0, 1$ is the Rydberg state occupation at site $j$, which clearly resolves the QPT. The domain wall density (DWD) is another indicator of the onset of the QPT [8]. Domain walls are defined as neighbouring Rydberg atoms in the same state or a Rydberg atom not in a Rydberg state on the open boundaries. The bottom pane of Fig. 5 shows the simulated DWD versus $\delta/\Omega$. The behaviour of $|M_s|$ and the DWD across the range of $\delta/\Omega$ values matches that from the experimental results in Figure 5 from Bernien *et al.* [8] extremely well.

Interestingly, depending on the cluster update type that is employed throughout these simulations, we observe drastically different autocorrelation times [41,42] for $|M_s|$. The right-hand pane of Fig. 5 shows the autocorrelation times for three different update procedures: performing line updates exclusively, performing a line update or a multibranch update with equal probabilities at every MC step, or performing multibranch updates exclusively. Each autocorrelation time curve shows a peak near the QPT, but the line update offers orders-of-magnitude better autocorrelation times compared to multibranch updates. Whether this critical slowing can be ameliorated further is a problem we leave for future work. Additionally, we see that introducing a non-zero $\varepsilon$ as mentioned in Sec. 3 has little effect on the actual performance of the algorithm. Although this may differ depending on $R_b$, $\delta/\Omega$, and system size, these results illustrate how choice of update (or combination of the updates) is crucial to simulation efficiency.

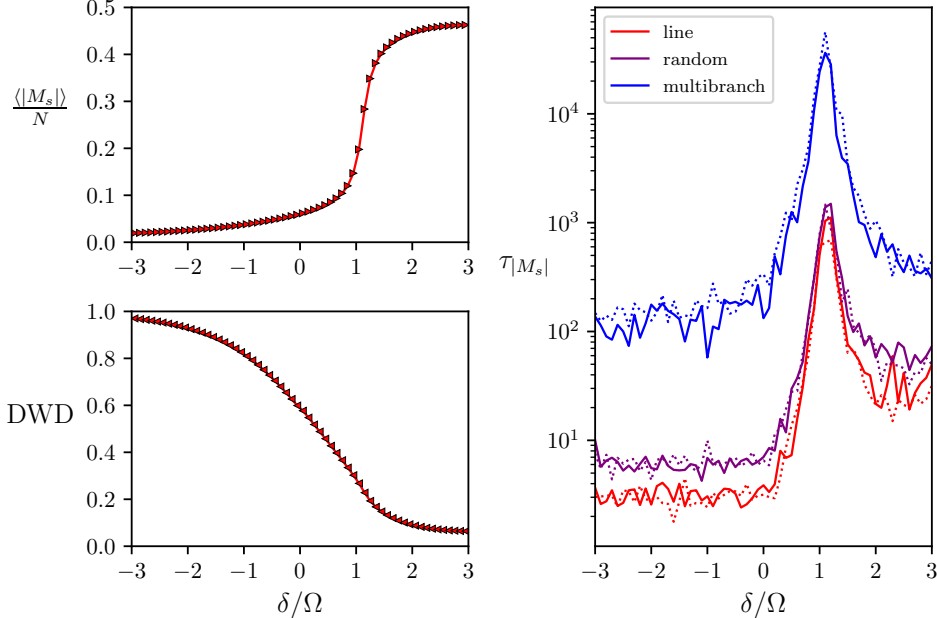

Figure 5: Absolute value of the staggered magnetization density $\langle|M_s|\rangle/N$ (top left), and the corresponding staggered magnetization autocorrelation times $\tau_{|M_s|}$ (right pane) for three different update procedures – line updates exclusively (red), randomly choosing line or multibranch updates at every MC step (purple), or multibranch updates exclusively (blue) – and different $\varepsilon$ values: $\varepsilon = 0$ (solid lines), or $\varepsilon = 0.1$ (dotted lines). The estimated DWD for an $N = 51$ 1D chain of Rydberg atoms with $R_b = 1.2$ as a function of $\delta/\Omega$ (bottom left). Each data point represents an independent SSE QMC simulation wherein $10^7$ successive measurements were taken and placed into 500 bins. These 500 binned measurements were then used to calculate statistics. Error bars for the plots on the left are smaller than the markers. A logarithmic binning analysis was performed on the full dataset to estimate the autocorrelation times.

## 4.2 256 atom 2D array

Next, we performed groundstate simulations of a $16 \times 16$ square lattice Rydberg array with open boundary conditions. We set $M = 10^5$, which we found gave sufficient energy convergence during preliminary runs. Independent simulations were performed over the range $\delta/\Omega \in [0, 1.75]$ in increments of 0.05, each performing $10^5$ equilibration steps followed by $10^6$ measurements.

For the value of $R_b = 1.2$, Samajdar *et al* reported the existence of a QPT from a disordered to checkerboard phase in two spatial dimensions on a square lattice [15]. The top left pane of Fig. 6 shows the absolute value of the staggered magnetization density where we observe this transition, and the top right pane shows the corresponding autocorrelation times [41,42] for exclusive multibranch updates, exclusive line updates, and randomly choosing between line and multibranch updates at every MC step. The orders-of-magnitude improvement in autocorrelation time when using line updates exclusively is apparent again for this system. Not only this, but the autocorrelation time for the multibranch curve does not show a peak near the transition into the checkerboard phase. This is most likely attributed to the fact that the staggered magnetization error bar sizes and non-monotonicity of the multibranch (blue) curve indicate non-ergodic behaviour.

Motivated by reported experimental results, Fig. 6 also shows the Rydberg excitation $\langle \hat{n} \rangle$, which shows good agreement in qualitative behaviour with the experimental results in Extended Data Figure 7 from Ebadi *et al.* [9], though this experimental data was extracted at a different value of $R_b$. Lastly, the autocorrelation time of the Rydberg excitation density is shown in the bottom right of Fig. 6, which again demonstrates that the line update's performance drastically exceeds that of the multibranch update in this parameter range.

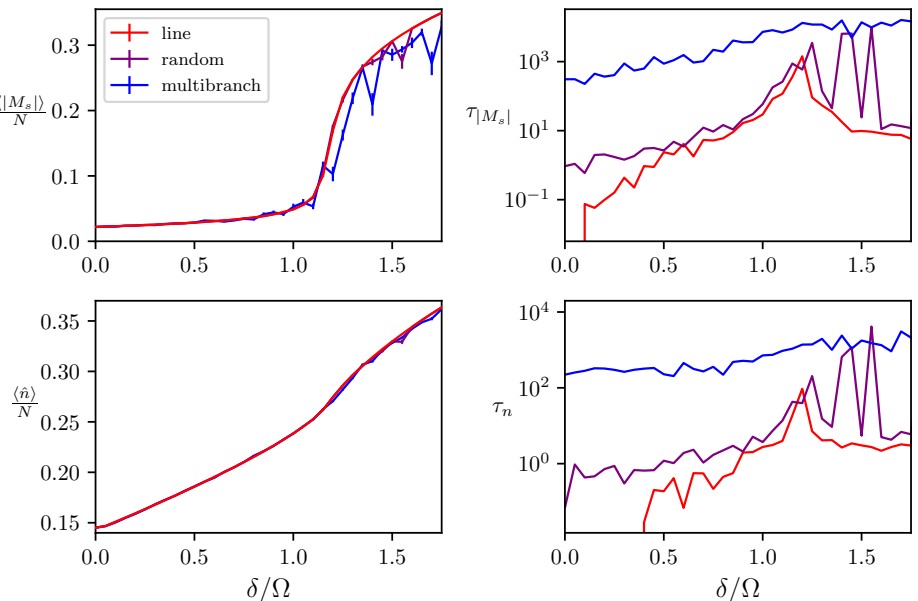

Figure 6: Absolute value of the staggered magnetization density $\langle |M_s| \rangle / N$ (top left), and the corresponding autocorrelation times $\tau_{|M_s|}$ (top right) for three different update procedures. The Rydberg excitation density and its autocorrelation time are plotted in the bottom row. Each data point represents an independent SSE QMC simulation of a 16×16 Rydberg array with $R_b = 1.2$, wherein $10^6$ successive measurements were taken and a logarithmic binning analysis was performed to estimate the autocorrelation times.

In order to further pin down exactly *why* the line update is so much more efficient than the multibranch update, we construct frequency histograms of both the counts and sizes of accepted and rejected clusters near the disordered-to-checkerboard phase transition. The cluster count histograms are constructed by counting the number of clusters in the simulation cell during each Monte Carlo step. Cluster size histograms are constructed similarly. In Fig. 7 we plot the relative frequencies of clusters against their sizes. First we must note that only certain cluster sizes are valid for each update; cluster sizes with frequency zero were not plotted. We see that the line update constructs clusters of more diverse sizes, with a gradual decay in frequency of larger clusters. On the other hand, the distribution of clusters constructed by the multibranch scheme is bimodal, with the dominant mode showing a very rapid decay with cluster size, followed by a smaller mode of very large rejected clusters. This indicates that the multibranch update tends to create a few very large clusters which will then rarely be flipped. Noting the distribution of rejected line clusters is much wider than that of the accepted clusters, it is clear that while the line update does build clusters of a greater variety of sizes, the larger clusters will not be flipped. We leave the possibility of a scheme which can flip larger clusters for future work.

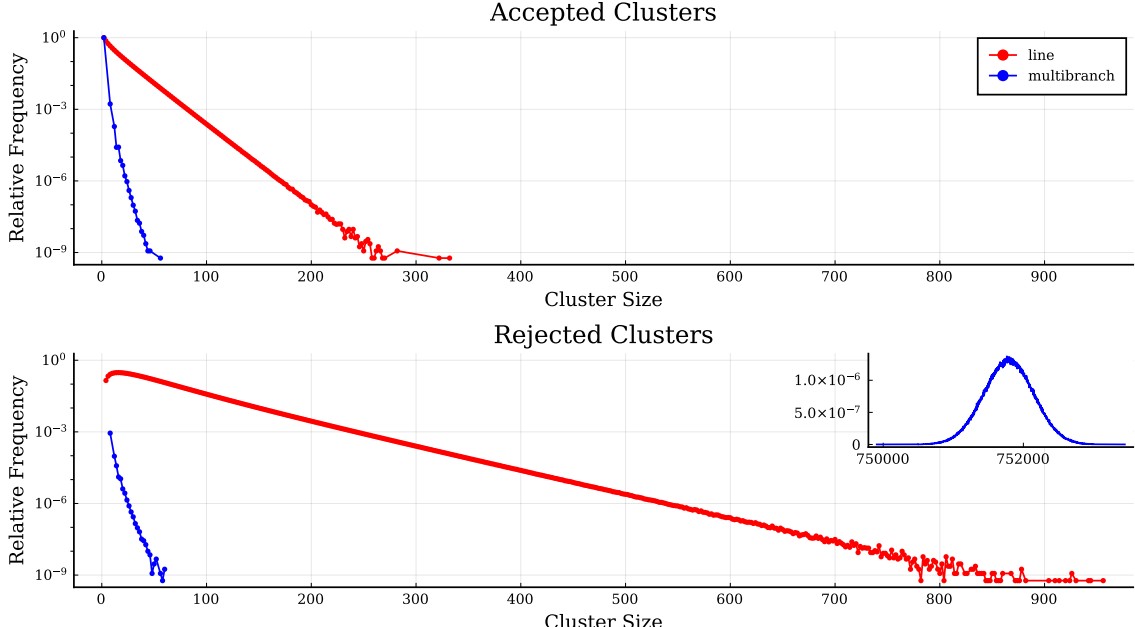

Figure 7: Cluster size histograms for the 2D Rydberg array at $R_b = 1.2$, $\delta/\Omega = 1.1$ for the two update types on a semi-log plot. Note that only certain cluster sizes are valid for each update; cluster sizes with frequency zero are not shown. Inset shows the second mode of the rejected cluster size histogram of the multibranch update on a linear plot.

In Fig. 8 we plot histograms of the number of clusters constructed in a single Monte Carlo step, as well as the mean cluster size, both as functions of the interaction truncation. Truncation was performed by eliminating interactions beyond the $k^{\text{th}}$ nearest-neighbour, where $k = \infty$ corresponds to no truncation. The cluster count histograms at each truncation approximately follow a Gaussian distribution, except for the rejected multibranch clusters which show a slight skew. We see that the multibranch update has a tendency to accept relatively few clusters in each Monte Carlo step while only rejecting a handful of clusters. Additionally, the mean cluster size shows that the accepted clusters constructed by the multibranch update are on average quite small (predominantly consisting of trivial clusters containing only two site operators) while the rejected clusters tend to grow quickly with interaction distance. In the case of the line update, increasing the truncation distance results in growth of both the accepted and rejected clusters, though the rejected clusters grow faster.[1] Combined with the data from Fig. 7, we can conclude that the multibranch update constructs a small number of very large clusters which will almost always be rejected. By breaking the clusters into smaller spatially-local slices, the line update is able to propose many more successful updates to the simulation cell.

---

[1]One may ask why the line update is sensitive to the interaction truncation in the first place as it is a spatially local update. While this is true, we must also keep in mind that more bond operators means there will on average be more bonds between two site operators in the SSE simulation cell, causing the temporal extent of the line clusters to grow.

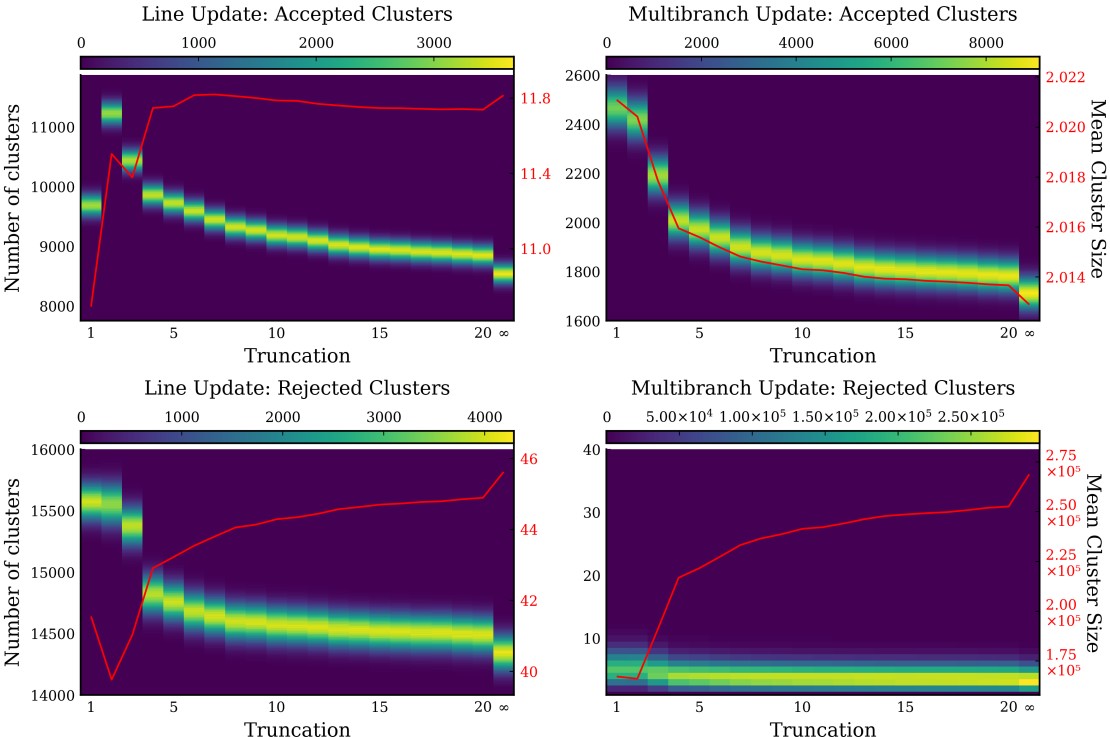

Figure 8: Frequency heatmap of cluster counts vs interaction truncation for the 2D Rydberg array at $R_b = 1.2$, $\delta/\Omega = 1.1$. The red line tracks the mean cluster size vs interaction truncation.

## 5 Conclusions

We have introduced a QMC algorithm within the SSE formalism that can efficiently simulate finite-temperature and ground state properties of Rydberg atom arrays in arbitrary dimensions. We have outlined the algorithm in both the finite-temperature and ground state projector formalism, emphasizing the theoretical frameworks as well as details required for practical implementation. In particular, we provide details of the Hamiltonian breakup into local operators, and introduce a modification of Sandvik's multibranch cluster update [32], suitable for Rydberg Hamiltonians with strong detuning. We also present an efficient estimator for the ground state energy, which is valid for any SSE algorithm containing an elementary operator that is a scalar multiple of the identity (including that for the transverse-field Ising model [32]).

In order to characterize the behaviour of the SSE algorithm, we study its efficiency in simulating recent results from experimental Rydberg arrays in one and two dimensions. In addition to convergence properties, we focus on Monte Carlo autocorrelation times for estimators of physical observables in the vicinity of quantum phase transitions which occur as a function of the detuning parameter. We compare in particular the original multibranch cluster update to a modified *line* update which is local in space but non-local in imaginary time. For some detunings near criticality, this new line update shows improvements of a least an order of magnitude in the autocorrelation time for some observables.

Our results show that this simple SSE QMC algorithm is very capable of simulating typical ground state observables measured in current state-of-the-art Rydberg array experiments. Considerable refinements of our algorithm are possible with straightforward modifications, including larger (plaquette) Hamiltonian breakups, and multicanonical sampling methods like

parallel tempering. These simulations will be able to offer more numerical insights into exotic physics contained in Rydberg atom arrays through detailed finite size scaling analyses, and will make available the wide array of well-developed SSE techniques, such as replica measurements of the Rényi entanglement entropies [37, 43–45].

Our SSE algorithm will be useful in directly characterizing equilibrium groundstate properties on Rydberg arrays of the exact size and lattice geometry of current experiments [8, 9, 46, 47]. In addition, QMC simulations such as this will be crucial for providing data for pretraining generative machine learning models, which are poised to become important tools in state reconstruction and tomography [48–51]. To this point, it is foreseeable that our SSE algorithm will be required to access system sizes beyond current experiments to facilitate the aforementioned numerical studies. We expect our SSE algorithm to set the standard for the performance of numerical simulation methods going forward. Finally, although the Rydberg Hamiltonian is fundamentally free of the sign problem – and hence lies in a complexity class where its groundstate properties are theoretically known to be amenable to efficient simulation – we have illustrated that devising an efficient algorithm is nontrivial in practice. The question we leave open is whether an efficient global SSE cluster update is available for all Rydberg interaction geometries which can be engineered in current and future experiments. Without algorithmic studies like the present to advance QMC and other simulation technologies forward [14, 15, 18, 52, 53], even sign-problem free Hamiltonians like those found in Rydberg arrays may stake a claim to experimental quantum advantage in the surprisingly near future.

## Acknowledgements

We thank S. Choi, M. Kalinowski, A. Sandvik, R. Samajdar, K. Slagle, R. Verresen and C. Zanoci for many stimulating discussions. We thank the Perimeter Institute for Theoretical Physics for the continuing support of PIQuIL.

**Author contributions**   All authors contributed equally to this work.

**Funding information**   This work was supported by the Natural Sciences and Engineering Research Council of Canada (NSERC), the Canada Research Chair (CRC) program, and the Perimeter Institute for Theoretical Physics. Research at Perimeter Institute is supported in part by the Government of Canada through the Department of Innovation, Science and Economic Development Canada and by the Province of Ontario through the Ministry of Colleges and Universities. Simulations were made possible by Compute Canada and the Shared Hierarchical Academic Research Computing Network (SHARCNET).

## A   Diagonal update probabilities

We will now derive the probabilities in Eqs. (16) and (17), which respectively dictate the removal or insertion of the diagonal operators $\hat{H}_{1,a}$ (Eq. (14c)) or $\hat{H}_{1,b}$ (Eq. (14d)) at a given imaginary time slice $p \in \{1, 2, \cdots, M\}$. It is worth emphasizing at this point that, though we often speak of inserting or removing "operators", the operator sequence $S_M$ is a sequence of operator *matrix elements*.

We begin with considering the insertion or removal of a *given* diagonal operator matrix element $\left\langle \alpha_{p-1} \middle| \hat{H}_{t_p,a_p} \middle| \alpha_p \right\rangle = \left\langle \alpha_p \middle| \hat{H}_{t_p,a_p} \middle| \alpha_p \right\rangle$ at the $p^{\text{th}}$ entry of the operator sequence $S_M$, and whose spatial location is $(i, j)$ (if $i = j$, this would correspond to a site operator akin to $\hat{H}_{1,a}$).

Specifically, we are restricting the potential diagonal operators to be inserted or removed to be only one operator whose matrix elements are uniform (i.e. do not depend on spatial location or $\{|\alpha_p\rangle\}$). We will discuss more complicated operator insertion or removal techniques that require choosing a diagonal operator from many options shortly.

Given the normalization constant in Eq. (7), the ratio of transition probabilities $P$ to insert or remove a given operator matrix element $\langle \alpha_p | \hat{H}_{t_p,a_p} | \alpha_p \rangle$ at the $p^{\text{th}}$ entry of the operator sequence $S_M$ must follow the detailed balance principle

$$\frac{P(S_M \rightarrow S'_M)}{P(S'_M \rightarrow S_M)} = \frac{\Phi(\{\alpha_p\}, S'_M)}{\Phi(\{\alpha_p\}, S_M)}, \tag{A.1}$$

where $\Phi$ is the generalized SSE configuration weight defined in Eq. (7), and $S'_M$ is the operator sequence after the insertion or removal update. In the case that the arbitrary operator matrix element $\langle \alpha_p | \hat{H}_{t_p,a_p} | \alpha_p \rangle$ was inserted ($n \rightarrow n+1$), the ratio of weights would be

$$\frac{\Phi(\{\alpha_p\}, S'_M)}{\Phi(\{\alpha_p\}, S_M)} = \frac{\beta^{n+1}(M-n-1)!}{M!} \left[ \prod_{p=1}^{M} \langle \alpha_{p-1} | \hat{H}_{t_p,a_p} | \alpha_p \rangle \right]_{[t_p,a_p]_p}$$

$$\times \frac{M!}{\beta^n(M-n)!} \left[ \prod_{p=1}^{M} \langle \alpha_{p-1} | \hat{H}_{t_p,a_p} | \alpha_p \rangle \right]^{-1}_{[0,0]_p}$$

$$= \frac{\beta \langle \alpha_p | \hat{H}_{t_p,a_p} | \alpha_p \rangle}{M-n},$$

where in the first line the subscripts on the products over $M$ denote that the $p^{\text{th}}$ element of the product contains the given operator type. Similarly, had the operator matrix element $\langle \alpha_p | \hat{H}_{t_p,a_p} | \alpha_p \rangle$ been removed ($n \rightarrow n-1$) and replaced with an identity element labelled by $[0,0]$, the ratio would be

$$\frac{\Phi(\{\alpha_p\}, S'_M)}{\Phi(\{\alpha_p\}, S_M)} = \frac{\beta^{n-1}(M-n+1)!}{M!} \left[ \prod_{p=1}^{M} \langle \alpha_{p-1} | \hat{H}_{t_p,a_p} | \alpha_p \rangle \right]_{[0,0]_p}$$

$$\times \frac{M!}{\beta^n(M-n)!} \left[ \prod_{p=1}^{M} \langle \alpha_{p-1} | \hat{H}_{t_p,a_p} | \alpha_p \rangle \right]^{-1}_{[t_p,a_p]_p}$$

$$= \frac{M-n+1}{\beta \langle \alpha_p | \hat{H}_{t_p,a_p} | \alpha_p \rangle}.$$

We will now discuss the specific dynamics of accepting a proposed transition to remove or insert a diagonal operator at a given imaginary time slice wherein there is more than one choice of operator while ensuring that the detailed balance principle (Eq. (A.1)) is enforced. However, we will still assume that the operator's spatial location is fixed.

## A.1   Metropolis scheme

A Metropolis-Hastings style update would require writing Eq. (A.1) as

$$\frac{\Phi(\{\alpha_p\}, S'_M)}{\Phi(\{\alpha_p\}, S_M)} = \frac{g(S_M \rightarrow S'_M)A(S_M \rightarrow S'_M)}{g(S'_M \rightarrow S_M)A(S'_M \rightarrow S_M)}, \tag{A.2}$$

where we've broken up the individual transition probabilities $P$ into a selection probability $g$ and an acceptance probability $A$. If we sample our diagonal operators – *not* their matrix elements – uniformly, the selection probabilities will cancel. Therefore, to satisfy the detailed balance principle we trivially require that

$$A(S_{M,p} \to S'_{M,p}) = \min\left(1, \frac{\Phi(\{\alpha_p\}, S'_{M,p})}{\Phi(\{\alpha_p\}, S_{M,p})}\right) = \min\left(1, \frac{\beta\left\langle\alpha_p\middle|\hat{H}_{t_p,a_p}\middle|\alpha_p\right\rangle}{M-n}\right), \qquad (A.3)$$

for an operator insertion, and

$$A(S_{M,p} \to S'_{M,p}) = \min\left(1, \frac{M-n+1}{\beta\left\langle\alpha_p\middle|\hat{H}_{t_p,a_p}\middle|\alpha_p\right\rangle}\right), \qquad (A.4)$$

for an operator removal.

It is well-known, however, that Metropolis-Hastings style updates are sub-optimal when there are more than two possible update choices. If we are to include choosing where diagonal operators are to be inserted spatially (i.e. inserting $\hat{H}_{1,b}$ requires a choice of spatial bond $(i,j)$) or we include multiple types of diagonal operators, it is preferable to instead use a heat-bath scheme.

## A.2 Heat-bath scheme

First, we will define operator matrix elements as

$$\Theta^{(i,j)}_{t_p,a_p}(\alpha_p) \equiv \left\langle\alpha_p\middle|\hat{H}^{(i,j)}_{t_p,a_p}\middle|\alpha_p\right\rangle, \qquad (A.5)$$

where the spatial dependence of the matrix element is given by physical indices $(i,j)$. For ease of notation, we will gather the labels $[t_p, a_p]$ and $(i,j)$ into one label $x$: $\Theta^{(i,j)}_{t_p,a_p}(\alpha_p) \equiv \Theta^{\alpha_p}_x$ In a heat-bath scheme, our transition probabilities are defined as

$$P(S_M \to S_M + \Theta^{\alpha_p}_x) = \frac{\Phi(\{\alpha_p\}, S_M + \Theta^{\alpha_p}_x)}{\Phi(\{\alpha_p\}, S_M) + \sum_{x',\alpha'}\Phi(\{\alpha_p\}, S_M + \Theta^{\alpha'}_{x'})}, \qquad (A.6)$$

where the sum in the denominator is over all diagonal operator matrix elements – excluding the identity – that can be inserted, and $S_M + \Theta^\alpha_x$ denotes inserting the operator matrix element (Eq. (A.5)) into the operator sequence $S_M$. It is straightforward to show that this satisfies the detailed balance condition of Eq. (A.1). However, the above equation must be simplified to enable efficient sampling; this is done by noting that all but one factor will divide out, giving

$$P(S_M \to S_M + \Theta^{\alpha_p}_x) = \frac{\beta\Theta^{\alpha_p}_x/(M-n)}{1 + \beta\sum_{x',\alpha'}\Theta^{\alpha'}_{x'}/(M-n)}, \qquad (A.7)$$

for an insertion (the numerator will be 1 for a removal). The diagonal update in this scheme will amount to constructing the discrete distribution above, sampling from it, and rejecting the insertion if the sampled matrix element did not match the actual state $\alpha_p$.

However, during the course of an SSE simulation, the expansion order $n$ fluctuates, forcing one to reconstruct this distribution every time we wish to insert a diagonal operator. Even with state-of-the-art sampling methods like the Alias method, reconstructing the distribution will cost $O(K)$ time, where $K$ is the number of diagonal operator matrix elements (i.e. the number of terms in the sum $\sum_{x',\alpha'}$). We may circumvent this costly overhead by using a hybrid approach which we call a "two-step" scheme.

### A.3 Two-step scheme

Beginning with the transition of inserting $(n \to n+1)$ any diagonal operator at the imaginary time slice $p$, the detailed balance condition reads

$$\frac{P(n \to n+1)}{P(n+1 \to n)} = \frac{\sum_{x,\alpha} \Phi(\{\alpha_p\}, S_M + \Theta_x^\alpha)}{\Phi(\{\alpha_p\}, S_M)} \, , \tag{A.8}$$

As there are only two options in this case (insert an operator or do not), we will use a Metropolis-Hastings style acceptance. When taking the ratio on the right-hand-side of Eq. (A.8), all factors in the weight $\Phi$ that do not belong to the specific imaginary time slice of interest will cancel. We are left with the following acceptance probability

$$A(n \to n+1) = \min\left(1, \frac{\beta \sum_{x,\alpha} \Theta_x^\alpha}{M-n}\right), \tag{A.9}$$

Similarly, for an operator removal,

$$A(n \to n-1) = \min\left(1, \frac{M-n+1}{\beta \sum_{x,\alpha} \Theta_x^\alpha}\right). \tag{A.10}$$

If the move to insert a diagonal operator is accepted, the choice of which diagonal operator to insert must still be made. We then perform a heat-bath step by sampling from the distribution

$$P(x, \alpha) = \frac{\Theta_x^\alpha}{\sum_{\alpha', x'} \Theta_{x'}^{\alpha'}} \, , \tag{A.11}$$

which can be done efficiently via the Alias method. The insertion is rejected if $\alpha \neq \alpha_p$. We now only need to initialize this distribution once at the beginning of our QMC simulation, and can sample from it in $O(1)$ time at every diagonal update step.

### A.4 Reducing rejections

There is, however, one last issue remaining. Since we are sampling from the set of all operator matrix elements, most of our sampled elements will be rejected as they often will not match the propagated state $\alpha_p$. Upon inspection, the distribution in Eq. (A.11) can be factorized into a distribution $P(x)$ and a conditional distribution $P(\alpha|x)$.

$$P(x, \alpha) = \frac{\Theta_x^\alpha}{\sum_{x', \alpha'} \Theta_{x'}^{\alpha'}} \, , \tag{A.12}$$

$$P(x)P(\alpha|x) = \frac{\sum_{\alpha'} \Theta_x^{\alpha'}}{\sum_{x', \alpha'} \Theta_{x'}^{\alpha'}} \times \frac{\Theta_x^\alpha}{\sum_{\alpha'} \Theta_x^{\alpha'}} \, . \tag{A.13}$$

To sample this and take the propagated state $\alpha_p$ into account, we first sample an operator from $P(x)$, then accept the proper matrix element with probability $P(\alpha_p|x)$. This is equivalent to the method discussed with the definition of Eq. (A.11), but it gives a clearer insight into the matrix element acceptance rate: $P(\alpha_p|x)$, which is the quantity we seek to maximize.

In order to improve our acceptance rate, we ask if it is necessary that $P(x) \sim \sum_\alpha \Theta_x^\alpha$, or if another function of the matrix elements will suffice. Call this unknown function $\Theta_x$. We consider the rate $P(\alpha_p|x) = \Theta_x^{\alpha_p}/\Theta_x$ averaged over the relevant marginal distribution of the

QMC configuration space, $P(\alpha_p)$:

$$
\begin{aligned}
\left\langle P(\alpha_p|x) \right\rangle_{\alpha_p} &= \frac{1}{\Theta_x} \sum_{\alpha_p} P(\alpha_p)\Theta_x^{\alpha_p} \\
&\leq \frac{(\max_\alpha \Theta_x^\alpha)}{\Theta_x} \sum_{\alpha_p} P(\alpha_p) \\
&= \frac{(\max_\alpha \Theta_x^\alpha)}{\Theta_x} .
\end{aligned}
$$

This ratio will always be less than one unless there is only one non-zero matrix element in a given diagonal operator $\hat{H}_{t,a}$. To maximize this ratio, we set $\Theta_x = \max_\alpha \Theta_x^\alpha$. Then, the heat-bath step now involves sampling the distribution

$$
P(x) = \frac{\max_\alpha \Theta_x^\alpha}{\sum_{x'}(\max_{\alpha'} \Theta_{x'}^{\alpha'})} , \tag{A.14}
$$

and then inserting the relevant matrix element with probability

$$
P(\alpha_p|x) = \frac{\Theta_x^{\alpha_p}}{\max_\alpha \Theta_x^\alpha} . \tag{A.15}
$$

For operators such as $\hat{H}_{1,a}$ where all matrix elements are equal, the acceptance ratio is always unity, hence this step can be skipped entirely. One may ask why we do not set $\Theta_x = \min_\alpha \Theta_x^\alpha$ as this would give a constant acceptance rate of unity. While this is true, we would no longer have a mechanism to enforce the relative occurrences of the different matrix elements in the simulation cell to be in line with their weights.

Lastly, we need to modify the insertion and removal probabilities, which now read

$$
A(n \to n+1) = \min\left(1, \frac{\beta\left[\sum_x(\max_\alpha \Theta_x^\alpha)\right]}{M-n}\right) = \min\left(1, \frac{\beta\mathcal{N}}{M-n}\right), \tag{A.16}
$$

and

$$
A(n \to n-1) = \min\left(1, \frac{M-n+1}{\beta\left[\sum_x(\max_\alpha \Theta_x^\alpha)\right]}\right) = \min\left(1, \frac{M-n+1}{\beta\mathcal{N}}\right). \tag{A.17}
$$

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
