# Peer review of "Stochastic Series Expansion Quantum Monte Carlo for Rydberg Arrays"

_SciPost Physics, doi:SciPost Phys. Core 7, 016 (2024)_

## Round 1 · Referee Report · Anonymous · 2021-8-13

Report

Arrays of Rydberg atoms are a powerful platform to realize strongly-interacting quantum many-body systems. It can be translated to an effective transverse field Ising model (TFIM) with long range interactions and vertical field, without sign problem. In this work, the authors develop a cluster scheme for this model. This work is an improvement of a previous SSE algorithm of normal TFIM.

I think this work sounds interesting and worthy to be published. However, I still have a question and hope authors can answer it. A similar algorithm for TFIM with a vertical field is also used in the supplementary materials of Phys. Rev. B 103, 104416 (2021). In these methods, the weights before/after flipping cluster are not equal, so I think you have to create the cluster first and calculate its weight to get the accept-probability. If you accept this update, then you can replace the configuration memory, else you have to keep the old arrays of operator string and spins. I think this step will cost more time than normal TFIM with accept-probability 0.5. Because in normal TFIM case, people can judge accept or not first (probability is independent on cluster), then update related arrays. I doubt it makes the computation complexity of this algorithm is no longer \beta*size as in normal SSE. I am just interested in whether you have good way to deal with this problem?

---

## Round 1 · Referee Report · Anonymous · 2021-8-13

Strengths

1- clear description of the algorithm
2- quality of the numerics
3- connection to experimental results

Weaknesses

1- presentation of the limitations of the algorithm

Report

This paper extends quantum Monte Carlo methods previously used for the transverse-field Ising model to a Hamiltonian describing an array of Rydberg atoms. In contrast to the pure transverse-field Ising model, this model lacks a spin-flip symmetry so that the conventional cluster updates change the configuration weight. The authors remedy this by introducing an a-posteriori acceptance probability for each cluster flip and find that the construction of space-local clusters improves ergodicity. Further, an efficient estimator for the ground state energy in the projector algorithm is derived.

As a proof of concept, one and two-dimensional systems are simulated close to the checkerboard quantum phase transition.

These results are of interest and should inspire further investigations given the recent interest in Rydberg systems. The paper is also otherwise clear and well written, up to some concerns detailed below. After these concerns are addressed, I think that the paper satisfies all criteria for publication in SciPost Physics.

Requested changes

1- The system sizes considered are small compared to what is possible e.g. in Ref. [30]. In the current manuscript, the reason for this is not clear. Refs. [8,9] suggest the choices $N=51$ and $N=256$ were inspired by experimental setups. This should be clearly stated in the main text. Is $N=256$ already the limit of what is possible? Are the autocorrelation times the bottleneck for this or something else?

2- The authors present data for a one-dimensional model. Due to the method focus of this paper, they should comment on the applicability of DMRG in this case. In particular, in Refs. [14-16], DMRG is already used to study Rydberg systems. (Why) should we use QMC in one dimension at $R_b = 1.2$?

3- On p.11 there is the statement “the operators in the sequence $S_M$ will most likely be dominated by one of $\{W_{i,j}^{(1)}, W_{i,j}^{(4)}\}$”. Is this really the case? In the checkerboard phase, I would naively expect that operators with differing states on their legs ($W^{(2)}$ and $W^{(3)}$) proliferate. In the shown example (Fig. 3), $W^{(3)}$ also has the highest weight out of all four. Please correct me if I am wrong.

4- Before introducing the ground state energy estimator in Sec. 3.3, it would be helpful to reference the already available ground state estimators such as the overlap method detailed in [A. W. Sandvik, PRL 95, 207203]. Does the new estimator have advantages over existing methods?

5- In the discussion of the autocorrelation times in the two-dimensional model (top of p. 17), there is an interesting point about normalizing autocorrelation times by the number of clusters formed. What exactly is meant by the “number clusters formed”? Does this number count the number of attempted cluster flips or the number of actually flipped clusters? Please clarify.
I would expect the actual size of the cluster in terms of flipped operators to play a role as well. Is there something one can say about the scaling of the acceptance probability for the multibranch and the line updates? My guess would be exponential decay in the number of operators per cluster. Then the line update would work better because its line clusters contain fewer operators.

6- Have the authors considered adding a constant $\epsilon$ to $C_{ij} = |\min(\dots)| + \epsilon$? This is routinely done in the context of the SSE directed loop update (e.g. Ref. [26]) and sometimes improves ergodicity by avoiding such W=0 situations. I doubt that this will fundamentally cure the autocorrelation issues, so there is no need to redo simulations. Mentioning the possibility of such an offset may nevertheless be helpful to the reader.

7- In the discussion of the results, a quantum phase transition and the checkerboard phase is mentioned. This should be shortly introduced to give the reader some overview over what is being studied.

8- currently, in the Figures, $\delta/\Omega$ is given as a ratio of $\Omega=1$ but $E$ and $T$ in Fig. 4 are not. The authors may consider to make it consistent by writing $E/Ω$ and $T/Ω$.

9- on page 15, the inline equation for $|M_s|$ enters into the margin. This should be fixed.

---

## Round 1 · Referee Report · Pranay Patil · 2021-8-22

Strengths

1. Provides a detailed explanation of the formulation of the stochastic series expansion quantum Monte Carlo (SSEQMC) method for currently relevant Rydberg atom arrays.
2. Step by step implementation protocol for the algorithm is presented for both finite temperature and the zero temperature projection method, along with the energy estimator, which can be easily generalized to other similar systems.
3. The line cluster algorithm proposed in this manuscript is shown to perform significantly better than a naive cluster algorithm, especially for the phase space of interest in the Rydberg atom arrays.
4. The appendices describe in detail the calculations of the probabilities involved in the Monte Carlo simulation, and further optimizations to improve performance. As this can also be easily extended to more general systems, practitioners of this method will benefit directly from these explanations.

Weaknesses

1. As the main focus of this manuscript is the line cluster update, it is essential to note that the expected range of parameters in which the update is expected to lead to efficient sampling is relatively small, and depends crucially on the values of W^{(1)-(4)}. The region of parameter space which corresponds to long line clusters is expected to suffer from small acceptance probabilities (this effect does not show up in a significant way at the system sizes considered here).
2. The performance of the line cluster is strongly tied to the region of parameter space being sampled, and thus is not naively expected to generalize to other Ising-like systems with interactions which break Z_2.

Report

The authors have presented the implementation of a quantum Monte Carlo method for Hamiltonians describing Rydberg atom arrays. The central contribution of this manuscript is the development of a line cluster method which performs significantly better than a naive cluster method for the parameter range of interest.
All the general acceptance criteria are met by this manuscript. From the list of expectations provided by the journal, this manuscript satisfies "Open a new pathway in an existing or a new research direction, with clear potential for multipronged follow-up work". One must note however that such line cluster methods have been used (arXiv:1812.05326,2008.11206), without a study of the improvement in ergodicity. The current manuscript fills this gap and holds promise for further improvements and developments in sophisticated cluster methods.
Typo: On page 5, above equation 7, I think you mean "operator sequence is less than 80%"

Requested changes

1. As far as cluster methods are concerned, one of the first checks for efficiency is usually performed through an examination of the success rate of proposed moves. This quantity can be easily extracted from simulations with small error bars and usually proves to be much less noisy than auto-correlation times. The manuscript will benefit from a short discussion of the behavior of the success rate as a function of Hamiltonian paramters.
2. (Optional (up to the authors' discretion); may prove to be too computationally involved to add to the current manuscript) Long line clusters are invariably bound to fail due to the large number of two-site operators which contribute to them (especially away from the special point in parameter space where the values of W^i are favorable). An alternative to this strategy would be to allow a Swendsen-Wang style cluster building algorithm, where operators are added to the cluster using a probability calculated from the different matrix elements of the operator. Something along these lines is suggested in arXiv:2009.03249 for a quantum clock model, and was found to work better than naive cluster of line updates. It would be intriguing to try to extend the same to the Hamiltonian studied in the current manuscript, and may lead to improvements in ergodicity.

---

## Round 1 · Referee Report · Natalia Chepiga · 2021-9-2

Strengths

- Detailed description of the algorithm
- Finite-temperature calculations are interesting and could compliment previous studies
- The paper provides a MonteCalro alternative to the existing ED- and tensor network-based numerical tools usually used for Rydberg atoms

Weaknesses

- The reference list in the introduction is a bit outdated (see the report)
- The manuscript does not report new physics
- The limitations of the algorithm have not been analyzed
- No comparison to alternative numerical methods previously used for Rydberg atoms (MPS, iDMRG, constrained DMRG, ED)

Report

The paper is purely methodological: it does not report new phenomena but introduces the numerical tool capable to reproduce the results obtained in the experiments. As for the methodological paper it however lacks a few essential ingredients: First, the manuscript does not provide a careful analysis of the limitations in terms of computational costs, systems sizes, accuracy etc. The authors limited themselves to system sizes available in experiments already a few years ago. Since the experiments on Rydberg atoms are progressing fast, in particular, in the number of trapped atoms, it would be interesting to have more systematic analysis of the limitations of the method in terms of reachable system sizes and other parameters that would put the method in a perspective. Second, I was lacking a comparison in terms of an accuracy and computational costs with respect to other available methods reporting results also on much larger system sizes (e.g. MPS results from the original Nature papers; iDMRG from Rader&Lauchli arXiv:1908.02068; and from Guidici et al. Phys. Rev. B 99, 094434; constrained DMRG from SciPost Phys. 6, 033 (2019)).

In addition, I would like to point out, that references in the introduction list some early papers and refer to the phase diagrams that have been recently corrected with more advanced numerical simulations in:
Phys. Rev. Lett. 122, 017205 (2019)
Phys. Rev. B 99, 094434 (2019)
Phys. Rev. Research 3, 023049 (2021)
And there are also relevant unpublished works:
Rader&Lauchli: arXiv:1908.02068
Chan et al: https://www.youtube.com/watch?v=bptOdSHo2dI&t=1664s

Finally, note, that even for the specific set of parameters chosen in the paper (\Omega=1; R_b=1.2) in 1D there is still an open question: the transition to the ordered Z_2 phase could either be 1st order or Ising and to the best of my knowledge the two possibilities have not been resolved yet for the van der Waals potential. By the way, Ref.14 at the bottom of p.14 is quite misleading since it refers to another transition out of Z_3 phase that takes place at larger values of R_b>2.

To summarize, the manuscript describes numerical tool complimentary to the exiting numerical methods used for Rydberg atoms and undoubtedly deserves to be published. However, the paper in its current form misses a few essential methodological ingredients that has to be addressed before the manuscript can be accepted for publication in SciPost.

Requested changes

See the report

---

## Round 2 · Referee Report · Anonymous (Referee 2) · 2023-2-5

Report

Considering the fast development and huge attention on the area of Rydberg array, I maintain my opinion that the paper is worthy to be published on the SciPost Physics.

However, I think the authors should demonstrate or modify their claim that the computation complexity of this algorithm is still \beta \times size. Although I in pricinple agree that "running updating cluster one after the other only results in the addition of a modest multiplicative constant, c, to the computational complexity of the cluster update step", it seems to introduce a "replace memory" process extrally. Because we have to prepare a copied array from the original array to store the information of a updating-cluster, there will introduce two more steps: if the update is accepted, the original array should be replace by the copied one; if rejected, the copied one should be recovered as the original one and begin the next step. Both these two processes introduce operations with a time complexity of roughly the cluster size. Assuming the average cluster size is proportional to the beta times system size, it would seem to result in an increase in complexity that is not a constant multiple.

I think it is better to delete the claim about the computation complexity below Eq. 21 or demonstrate it via numeric results. After the question has been properly treated, I think the manuscript can be recommendated for publication.

---

## Round 2 · Referee Report · Natalia Chepiga (Referee 4) · 2023-2-10

Report

I acknowledge the authors' decision to leave comparison with other numerical techniques outside of the scope of the manuscript and that limitation of the algorithm " will be left for future studies".
I find that the manuscript does not fulfill the SciPost Physics acceptance criteria (groundbreaking discovery/breakthrough/new research direction/synergetic link between different research areas). I therefore recommend it for publication in SciPostPhysicsCore.

---

## Round 2 · Referee Report · Anonymous (Referee 1) · 2023-2-13

Report

The authors have successfully addressed the technical remarks of the previous round, adding clarifications and additional data that improve the overall quality of the manuscript.

The determination of the limits of applicability of the presented method and a more thorough discussion of the competitiveness with tensor network approaches have been left for future studies.

However, these elements are crucial in order to demonstrate a groundbreaking discovery or breakthrough, as required by the SciPost Physics acceptance criteria.

Therefore, I recommend the manuscript in its current state for publication in SciPost Physics Core.

---

## Round 2 · Referee Report · Pranay Patil (Referee 3) · 2023-2-13

Report

The authors have made appropriate modifications to the manuscript, and answered the questions raised in the previous report. The generality of the proposed update still remains a question as the authors always work in the regime where the Rydberg interaction and transverse field strength are comparable ($R_b$=1.2 and $\Omega=1$). One would expect that upon increasing $R_b$ (enforcing the Rydberg blockade more rigorously), the success probability of the line cluster update reduces exponentially in $R_b/\Omega$ ($C_{ij}$ approximated to $R_b$ for large $R_b$), similar to metropolis flips which are suppressed exponentially at low temperatures in classical spin system simulations (the line cluster update is a single site operation for real space). Given that a majority of theoretical proposals are formulated in the manifold created by enforcing the Rydberg blockade and mapping to dimer models, the line cluster update appears to have limited utility. This is also consistent with the two cases which the authors have studied (linear and square lattice), both of which are unfrustrated and do not require careful worm updates to simulate the kind of physics expected in spin liquids. Due to the above argument, I would recommend that this manuscript is suitable for SciPost Physics Core.

---

## Round 2 · Author Response

Other changes:

Minor notational and linguistic changes were made to the Appendix. We also add two sentences after Eq. 40 to explain why an alternative form of the operator weight, $\Theta_x$, would not be useful.

Response to Anonymous Report 1:

We thank the referee for their rigorous assessment of our work. We will address each of the requested changes in the sequence they were originally reported in.

1. The system sizes considered are small compared to what is possible e.g. in Ref. [30]. In the current manuscript, the reason for this is not clear. Refs. [8,9] suggest the choices N=51 and N=256 were inspired by experimental setups. This should be clearly stated in the main text. Is N=256 already the limit of what is possible? Are the autocorrelation times the bottleneck for this or something else?

We deliberately chose the N=51 and N=256 system sizes for two reasons. The first, as was mentioned by the referee, is the inspiration drawn from the experimental setups. The second is for practical purposes pertaining to the length and content of the paper: our aim is to demonstrate how the algorithm works as clearly and concisely as possible in a manuscript of limited length. Note, N=256 is not the limit of what is possible; that limit is dependent on the specific context and will be left for future studies.

At the beginning of Section 4 (Results), we already included a paragraph showcasing the motivation for our chosen results. We have added a sentence in the last paragraph of Section 5 (Conclusions) to clarify that the performance ceiling of our algorithms is yet to be determined.

2. The authors present data for a one-dimensional model. Due to the method focus of this paper, they should comment on the applicability of DMRG in this case. In particular, in Refs. [14-16], DMRG is already used to study Rydberg systems. (Why) should we use QMC in one dimension at Rb=1.2?

DMRG and QMC are different and complementary tools, and the choice to use one or the other depends on context. Indeed, DMRG is extremely useful for one-dimensional models and should be used when possible. It may however be limited in its capacity to faithfully capture the effects of long-range interactions, important in the present case. Note, in Refs. [15,16] there is some form of truncation on the all-to-all interactions in the Rydberg Hamiltonian. Therefore, in order to faithfully simulate experiments wherein all-to-all interactions are a reality, one may wish to use QMC, even in one dimension (regardless of the choice of the blockade radius, Rb). Detailed comparisons of QMC and DMRG are however left for future studies.

3. On p.11 there is the statement “the operators in the sequence SM will most likely be dominated by one of {W(1)i,j,W(4)i,j}”. Is this really the case? In the checkerboard phase, I would naively expect that operators with differing states on their legs (W(2) and W(3)) proliferate. In the shown example (Fig. 3), W(3) also has the highest weight out of all four. Please correct me if I am wrong.

Indeed, the referenced quote is not correct; we thank the referee for pointing out this inconsistency. We have reworked the argument to instead compare the combinatorics of multibranch clusters vs line clusters. We believe this to be an overall cleaner argument. Additionally, we removed the referenced figure as it is now superfluous.

4. Before introducing the ground state energy estimator in Sec. 3.3, it would be helpful to reference the already available ground state estimators such as the overlap method detailed in [A. W. Sandvik, PRL 95, 207203]. Does the new estimator have advantages over existing methods?

Estimators, besides those related to ones that can be calculated based on computational-bases samples (e.g. the magnetization, staggered magnetization, etc.), in the projector / ground state SSE formalism are unique to the specific SSE algorithm. The estimator in [A. W. Sandvik, PRL 95, 207203] would not be applicable in the case of our SSE algorithm because the entire SSE formalism is different (the computational basis, the Hamiltonian, and how the Hamiltonian is broken into local parts are all different). Therefore, our estimator has the distinct advantage in that it is applicable to SSE algorithms wherein one of the defined local parts of the Hamiltonian of interest is a multiple of the identity.

Note also, the overlap method requires estimating the diagonal part of the Hamiltonian. Since the current Hamiltonian involves all-to-all interactions, each sample would require O(N^2 + M) time, where N is the number of sites and M is the projection length. Our method only involves counting the number of single-site diagonal operators in the simulation cell, a process which takes only O(M) time per sample.

5. In the discussion of the autocorrelation times in the two-dimensional model (top of p. 17), there is an interesting point about normalizing autocorrelation times by the number of clusters formed. What exactly is meant by the “number clusters formed”? Does this number count the number of attempted cluster flips or the number of actually flipped clusters? Please clarify. I would expect the actual size of the cluster in terms of flipped operators to play a role as well. Is there something one can say about the scaling of the acceptance probability for the multibranch and the line updates? My guess would be exponential decay in the number of operators per cluster. Then the line update would work better because its line clusters contain fewer operators.

We thank the reviewer for pointing out the subtleties involved in the referenced point. We've decided to remove that point since a rigorous discussion would likely be too lengthy and is outside the scope of the paper.

6. Have the authors considered adding a constant ϵ to Cij=|min(…)|+ϵ? This is routinely done in the context of the SSE directed loop update (e.g. Ref. [26]) and sometimes improves ergodicity by avoiding such W=0 situations. I doubt that this will fundamentally cure the autocorrelation issues, so there is no need to redo simulations. Mentioning the possibility of such an offset may nevertheless be helpful to the reader.

We have added a few sentences to the paragraph following Eqs. 14a-d. Additionally, we redo our correlation time plot for the 1D chain (Fig 5) to include non-zero epsilon simulation data. Note, however, that we have defined epsilon slightly differently than is typical for SSE; by defining it as a multiplicative constant we avoid the risk of an additive epsilon "washing out" the finer structure of a weak bond operator. As we discuss in the relevant section of the paper, a non-zero epsilon does not change the general behaviour of the correlation time, which is determined predominantly by the choice of algorithm. It should be noted that preliminary runs using an additive epsilon gave similar results.

7. In the discussion of the results, a quantum phase transition and the checkerboard phase is mentioned. This should be shortly introduced to give the reader some overview over what is being studied.

We have added a sentence at the beginning of the second paragraph in Section 4.2 with a reference to the study showing the existence of such a transition.

8. Currently, in the Figures, δ/Ω is given as a ratio of Ω=1 but E and T in Fig. 4 are not. The authors may consider to make it consistent by writing E/Ω and T/Ω.

We have changed the relevant figures accordingly.

9. on page 15, the inline equation for |Ms| enters into the margin. This should be fixed.

We have changed this equation to stand alone rather than be inlined.

Response to Anonymous Report 2:

We thank the referee for their assessment of our work. We will address their concern in the below.

I think this work sounds interesting and worthy to be published. However, I still have a question and hope authors can answer it. A similar algorithm for TFIM with a vertical field is also used in the supplementary materials of Phys. Rev. B 103, 104416 (2021). In these methods, the weights before/after flipping cluster are not equal, so I think you have to create the cluster first and calculate its weight to get the accept-probability. If you accept this update, then you can replace the configuration memory, else you have to keep the old arrays of operator string and spins. I think this step will cost more time than normal TFIM with accept-probability 0.5. Because in normal TFIM case, people can judge accept or not first (probability is independent on cluster), then update related arrays. I doubt it makes the computation complexity of this algorithm is no longer \beta*size as in normal SSE. I am just interested in whether you have good way to deal with this problem?

While it is true that one can fuse the cluster identification and flipping loops in the normal TFIM algorithm, running them one after the other only results in the addition of a modest multiplicative constant, c, to the computational complexity of the cluster update step. It is straightforward to show that 1 <= c <= 2 as c is simply the average cluster flipping rate plus 1 (because we only run the flipping loop if the cluster flip has been accepted). Unfortunately, we have not yet found an efficient method for fusing the cluster identification and flipping loops (not for lack of trying!) and are uncertain whether such an optimization is possible. We have added two sentences to the paragraph below Eq. 21 to clarify this.

Response to Report 3 from Dr. Patil:

We thank Dr. Patil for their assessment of our work. We will address concerns in the below.

1. As far as cluster methods are concerned, one of the first checks for efficiency is usually performed through an examination of the success rate of proposed moves. This quantity can be easily extracted from simulations with small error bars and usually proves to be much less noisy than auto-correlation times. The manuscript will benefit from a short discussion of the behavior of the success rate as a function of Hamiltonian paramters.

We thank the referee for this suggestion. We have examined the acceptance rates of both the line and multibranch updates. Counter-intuitively, the multibranch update gave an acceptance rate very close to one! Further investigation reveals this to be an artifact of averaging a highly skewed distribution. The multibranch update (for the Rydberg Hamiltonian) has a tendency to build a handful of extremely large clusters, which are almost never flipped, along with a large number of "trivial clusters", which are always flipped. We add a discussion of the distributions of accepted and rejected cluster sizes and counts for both updates in section 4.2, as well as new Figures (7 & 8).

2. (Optional (up to the authors' discretion); may prove to be too computationally involved to add to the current manuscript) Long line clusters are invariably bound to fail due to the large number of two-site operators which contribute to them (especially away from the special point in parameter space where the values of W^i are favorable). An alternative to this strategy would be to allow a Swendsen-Wang style cluster building algorithm, where operators are added to the cluster using a probability calculated from the different matrix elements of the operator. Something along these lines is suggested in arXiv:2009.03249 for a quantum clock model, and was found to work better than naive cluster of line updates. It would be intriguing to try to extend the same to the Hamiltonian studied in the current manuscript, and may lead to improvements in ergodicity.

We agree that this would be intriguing, however we leave it for future studies.

Response to Report 4 from Dr. Chepiga:

The paper is purely methodological: it does not report new phenomena but introduces the numerical tool capable to reproduce the results obtained in the experiments. As for the methodological paper it however lacks a few essential ingredients: First, the manuscript does not provide a careful analysis of the limitations in terms of computational costs, systems sizes, accuracy etc. The authors limited themselves to system sizes available in experiments already a few years ago. Since the experiments on Rydberg atoms are progressing fast, in particular, in the number of trapped atoms, it would be interesting to have more systematic analysis of the limitations of the method in terms of reachable system sizes and other parameters that would put the method in a perspective.

We agree that such a thorough analysis would be interesting; however we believe that this is well out of the scope of the current manuscript (which already runs 25 pages long).

Second, I was lacking a comparison in terms of an accuracy and computational costs with respect to other available methods reporting results also on much larger system sizes (e.g. MPS results from the original Nature papers; iDMRG from Rader&Lauchli arXiv:1908.02068; and from Guidici et al. Phys. Rev. B 99, 094434; constrained DMRG from SciPost Phys. 6, 033 (2019)).

One generally expects MPS-based methods to be competitive with QMC in one dimension. However it is well documented that MPS/DMRG faces exponential scaling problems in two dimensions and higher. It is also known from a large body of previous research that QMC does not face similar problems. We agree with the referee that a comparison of QMC and DMRG is lacking and would be interesting, however there are clearly many variables that would have to be controlled to make such a comparison quantitative; dimension, interaction truncation, etc., making it clearly outside of the scope of our manuscript.

In addition, I would like to point out, that references in the introduction list some early papers and refer to the phase diagrams that have been recently corrected with more advanced numerical simulations in: Phys. Rev. Lett. 122, 017205 (2019) Phys. Rev. B 99, 094434 (2019) Phys. Rev. Research 3, 023049 (2021)

And there are also relevant unpublished works: Rader&Lauchli: arXiv:1908.02068

Chan et al: https://www.youtube.com/watch?v=bptOdSHo2dI&t=1664s

Finally, note, that even for the specific set of parameters chosen in the paper (\Omega=1; R_b=1.2) in 1D there is still an open question: the transition to the ordered Z_2 phase could either be 1st order or Ising and to the best of my knowledge the two possibilities have not been resolved yet for the van der Waals potential. By the way, Ref.14 at the bottom of p.14 is quite misleading since it refers to another transition out of Z_3 phase that takes place at larger values of R_b>2.

We have added citations to arXiv:1908.02068 as well as Phys. Rev. Research 3, 023049 (2021). The misleading reference to Ref.14 has been removed.

---

## Round 2 · List of Changes

• We have added a sentence in the last paragraph of Section 5 (Conclusions) to clarify that the performance ceiling of our algorithms is yet to be determined.
  • We have reworked the argument on the dominance of {W(1)i,j,W(4)i,j} terms to instead compare the combinatorics of multibranch clusters vs line clusters. Additionally, we removed the referenced figure as it is now superfluous.
  • We have removed a point on autocorrelation times in the two-dimensional model on the top of page 17.
  • We have added a few sentences to the paragraph following Eqs. 14a-d.
  • We have redone our correlation time plot for the 1D chain (Fig 5) to include non-zero epsilon simulation data.
  • We have added a sentence at the beginning of the second paragraph in Section 4.2 with a reference to a study.
  • All figures' axes are in units of \Omega where applicable.
  • The equation for |Ms| on page 15 has been fixed.
  • We have added two sentences to the paragraph below Eq. 21 .
  • We have added a discussion of the distributions of accepted and rejected cluster sizes and counts for both updates in section 4.2
  • New figures: Figure 7 and 8
  • We have added citations to arXiv:1908.02068 as well as Phys. Rev. Research 3, 023049 (2021).
  • The misleading reference to Ref. 14 has been removed.
  • Minor notational and linguistic changes have been made to the Appendix.
  • We have added two sentences after Eq. 40 to explain why an alternative form of the operator weight, \Theta_x, would not be useful.

---

## Editorial Decision

published